# Thermal and Sono—Aqueous Reforming of Alcohols for Sustainable Hydrogen Production

**DOI:** 10.3390/molecules29204867

**Published:** 2024-10-14

**Authors:** Choon Wee Kee, Jia’E Zheng, Wei Jie Yap, Roy Ou Yong, Yan Liu

**Affiliations:** Institute of Sustainability for Chemicals, Energy and Environment (ISCE^2^), Agency for Science, Technology and Research (A*STAR), 1 Pesek Road, Jurong Island, Singapore 627833, Singapore

**Keywords:** hydrogen, aqueous phase reforming, catalyst stability, ultrasound, alcohols, sustainability, sonolysis

## Abstract

Hydrogen is a clean-burning fuel with water as its only by-product, yet its widespread adoption is hampered by logistical challenges. Liquid organic hydrogen carriers, such as alcohols from sustainable sources, can be converted to hydrogen through aqueous-phase reforming (APR), a promising technology that bypasses the energy-intensive vaporization of feedstocks. However, the hydrothermal conditions of APR pose significant challenges to catalyst stability, which is crucial for its industrial deployment. This review focuses on the stability of catalysts in APR, particularly in sustaining hydrogen production over extended durations or multiple reaction cycles. Additionally, we explore the potential of ultrasound-assisted APR, where sonolysis enables hydrogen production without external heating. Although the technological readiness of ultrasound-assisted or -induced APR currently trails behind thermal APR, the development of catalysts optimized for ultrasound use may unlock new possibilities in the efficient hydrogen production from alcohols.

## 1. Introduction

Hydrogen, as a sustainable fuel, holds much promise as its combustion produces solely water, an environmentally benign product. However, it is well-documented that its key limitations for widespread uses include its low volumetric energy density due to its standard state being a gas, which is related to the logistic issues that are associated with transporting H_2_.

Alcohols that can be derived from biomass are promising candidates as liquid organic hydrogen carriers. The reforming of alcohols allows hydrogen to be produced by reacting the alcohols and water over a catalyst (Equation (1)). While the majority of the world’s hydrogen is produced by steam reforming, the feedstock is natural gas which is considered a non-renewable resource [1].

The steam-reforming (SR) process is carried out with water in its gaseous phase within the reactor, typically below the saturated vapor pressure of the water–alcohol mixture. SR of alcohols such as methanol and ethanol are mature technologies that have been applied on an industrial scale [2]. Selected reviews on SR are listed here. The use of nickel as a cost-effective active metal in SR of alcohols such as methanol, ethanol, and glycerol has been the focus of reviews [3,4]. Coke deposition is recognized as a main contributor to the deactivation of the catalyst in SR; Sharma et al. extensively discussed this in the context of ethanol SR [5]. Achomo et al. provided a holistic review on methanol SR which touched on thermodynamic, kinetic, catalysts (mainly Cu and Pd-based), and reactor aspects [6]. Yang et al. discuss the development of catalytic systems for methanol SR with a focus on energy–mass conversion, highlighting advancements in catalyst durability, carbon deposition resistance, and sintering resistance through material modification, additives, and structural optimization [7]. Sorption-enhanced steam reforming has been explored as a means to capture the CO_2_ formed during the reaction [8,9,10]. The reviews listed in this paragraph are by no means an exhaustive list of reviews on the SR of alcohols and readers are referred to the citations in these reviews to find those that preceded them.
(1)$$C_{n}H_{2n+2}O_{n}+nH_{2}O⇌nCO_{2}+\left(2n+1\right)H_{2}$$

The focus of this review will be on aqueous-phase reforming (APR) of alcohols with heterogeneous catalysts. APR involves the use of pressure above the saturated vapor pressure of the liquid to keep it in its liquid phase during the reforming reaction. This has been applied both in heterogeneous and homogeneous catalysis [11,12,13]. The commonly cited advantages of APR include allowing the reaction to be performed at a much lower temperature, and being less energy-intensive relative to SR as there is no need to vaporize the reactants. The ability to use a lower temperature is realized with feedstock such as ethanol and glycerol, the reforming of which is generally performed at a much higher temperature in SR than in APR [3,4]. For methanol, SR and APR reaction temperatures are generally comparable. The omission of the requirement for feedstock vaporization in APR enables it to be highly versatile with regard to the choice of feedstocks. Higher boiling points or non-volatile feedstocks such as sugars, polyols, cellulose, and amino acids have been explored [14,15]. Complex aqueous mixtures such as wastewater streams and glycerol from biodiesel production are sustainability-relevant feedstocks from which APR can potentially find industrial applications. In addition, aqueous inorganic bases can be used in APR to capture the CO_2_ formed in situ and modify the reaction profile.

The calculated saturated vapor pressures of model alcohols at common composition and temperature that are relevant to the APR reactions in this review are given in Table 1 [16,17]. Three different temperatures and two different weight percentages in water for each alcohol were considered. Due to the higher vapor pressure of methanol and ethanol, performing the APR at 250 °C with high alcohol content will require much higher pressures than with ethylene glycol and glycerol. However, it is important to note that when a carrier gas is used, as in the case of a fixed-bed reactor, the relative feed rate between the gas and liquid will influence the amount of liquid present during the continuous flow operation. Similarly, in a batch reactor, the ratio of headspace volume to liquid volume will determine whether the saturated vapor pressure can be reached, affecting the distribution of feedstock between the vapor and liquid phases.

A catalyst’s stability over an extended duration is highly relevant in an industrial setting. One of the recommended metrics of stability for industry—catalyst consumption (kg-cat per tons-product)—[18] has an ideal value for catalyst consumption of less than 0.1, which implies in this case, that for 1 g of catalyst, 10 kg (or about 5 kilomoles) of H_2_ has to be produced. The hydrothermal condition that a catalyst is subjected to in APR poses high stress on its stability [19]. Mechanisms for catalyst deactivation during a hydrothermal reaction are summarized in Figure 1. With the exception of poisoning, most of them are of relevance to APR. For a detailed discussion on these deactivation mechanisms, the reader is referred to authoritative reviews in the literature [20,21,22] and the references cited therein.

The objective of this review is to provide a semi-quantitative overview of stability in the aqueous-phase reforming of alcohols. Since hydrogen production is the focus of this review, activity will be measured in units of mmol-H_2_/g-cat/h whenever possible, and stability will be defined as the ability of the catalyst to maintain the activity on stream (in a fixed bed reactor) or upon recycling (in a batch reactor).

To avoid complications arising from complex aqueous mixtures, we will focus our discussion on model alcohols such as methanol, ethanol, ethylene glycol, and glycerol (Figure 2), except when strong stability data is available. These feedstocks, except ethanol, are commonly used in the literature when investigating the activity and stability of metals on a support. Among these model alcohols, glycerol is the most complex for aqueous-phase reforming (APR); theoretically, it can produce seven molecules of hydrogen per molecule of glycerol. However, catalysts and reaction conditions determine the experimental distribution of the various products listed in Figure 2. Methanol and ethylene glycol can theoretically produce three and five molecules of hydrogen per molecule, respectively. Ethanol, the least reported feedstock for APR, can produce only two molecules of hydrogen per molecule, as the methane formed cannot be reformed under APR reaction conditions.

The water–gas shift reaction (WGSR) (Figure 2) is generally accepted as the process that generates CO_2_ and hydrogen from CO and water in APR. The WGSR becomes thermodynamically more favorable as the temperature decreases [23]. It removes CO during aqueous-phase reforming (APR), resulting in a very low level of CO in the gaseous product. This is one of the advantages of APR, as the low level of CO, together with high pressure, makes the effluent amenable to further purification for fuel cell applications [24]. However, the reduction of CO or CO_2_ to methane can negatively impact hydrogen production. These side reactions become significant when metals such as Ni, Co, Ru, and Fe, which are highly active in methanation, are present, leading to the formation of a significant amount of CH_4_ [25].

Due to the promising nature of APR for hydrogen production, numerous reviews have been published over the years, each offering a different approach. The Dumesic group, pioneers in aqueous-phase reforming (APR), extensively reported on the APR of glycerol. Their review discussed APR from the perspectives of thermodynamics and kinetics [14]. Chen et al. reviewed APR, focusing on biomass-derived alcohols and emphasizing Pt- and Ni-based catalysts [26]. Coronado et al. extensively reviewed the catalytic APR of oxygenated hydrocarbons from biorefinery water fractions, highlighting Pt-based and Ni-based catalysts, operational challenges, and optimization strategies for hydrogen production efficiency [27]. Vaidya and Lopez-Sanchez emphasized recent advances in catalyst development and the potential for efficient hydrogen production with reduced CO levels [28]. Pipitone et al. conducted a comprehensive review of catalysts in APR from 2014–2020, providing crucial information on how various parameters in catalyst synthesis influence APR outcomes [29]. Azizan et al. addressed challenges in catalyst design and reactor engineering that must be overcome to make APR an industrially relevant process [30]. Tian et al. discussed APR in terms of different classes of feedstocks that can be derived from biomass [31]. Lastly, Joshi and Vaidya discussed advances in catalyst design, reactor engineering, thermodynamics, and kinetics of APR, as well as the coupling of different processes to APR, and provided various relevant techno-economic and life-cycle analysis examples [15]. These reviews collectively offer a detailed overview of the current state and future perspectives in APR technology, serving as valuable references for researchers and practitioners in the field.

In addition to the relatively well-studied thermal aqueous-phase reforming, the use of ultrasound has emerged as an activation method to achieve ambient condition reforming of alcohols. Ultrasound induces acoustic cavitation, involving the formation, growth, and collapse of gas bubbles. This process can create localized environments with extreme temperatures and pressures that last for sub-microsecond durations. Consequently, the reactions that occur are typically non-equilibrium and strongly depend on the gaseous content present during acoustic cavitation.

The rest of this review is divided into two parts. Section 2 focuses on thermally catalyzed aqueous-phase reforming (APR), organized by the type of support used in the catalyst. This approach is motivated by the limitation of material choice for the support due to their instability under the hydrothermal conditions present in thermal APR. In Section 2.1, we examine Al_2_O_3_-based catalysts and the improvements in stability achieved through support modification. Section 2.2 covers CeO_2_-supported catalysts, which enable APR at lower temperatures than typically reported. Section 2.3 discusses ZrO_2_ and related supports for highly stable APR catalysts. Other metal-supported catalysts are reviewed in Section 2.4. Carbon-based supports are explored in Section 2.5. Finally, Section 2.6 concludes the discussion on thermal APR with molybdenum-carbide-supported catalysts, which exhibit strong hydrogen production activity, and the relatively less-studied molybdenum sulfide. In Section 3, we shift our focus to ultrasonic-assisted or -enabled hydrogen production from aqueous mixtures of alcohols.

## 2. Discussion—Thermal Aqueous-Phase Reforming of Alcohols

Our discussion on the thermal aqueous-phase reforming (APR) of alcohols will be organized according to the supports that are used in the APR catalysts. Under conditions relevant to aqueous-phase reforming, the choice of support for a heterogeneous catalyst is rather limited (Figure 3). Carbon-based materials, ZrO_2_, and TiO_2_ are among those with excellent hydrothermal stability [32]. In contrast, silica- and alumina-based supports are best avoided due to their limited hydrothermal stability. Commonly cited mechanisms of deactivation in APR include leaching of active metals, sintering, support phase transition, and carbon deposition.

Pertinent to our discussion on support, Dumesic and co-workers evaluated platinum supported on a wide range of materials, including Al_2_O_3_, carbon, CeO_2_, SiO_2_, SiO_2_-Al_2_O_3_, TiO_2_, and ZnO [33]. They observed that supports like SiO_2_ and CeO_2_ tend to dissolve or disintegrate in the reforming environment, making them less suitable for prolonged use. This was evidenced by the detection of Si and Ce traces in the reactor effluent from the Pt/SiO_2_ and Pt/CeO_2_ catalysts, respectively, after only 6 h on stream at 210 °C (483 K), as analyzed by Inductively Coupled Plasma Atomic Absorption Spectroscopy (ICP-AAS).

With these challenges in mind, we now turn our attention to Al_2_O_3_ as a support material, which will be discussed in Section 2.1. Al_2_O_3_ demonstrated superior resistance to leaching, with less than 1 ppm of Al found after 24 h on stream at 225 °C (498 K) [33], indicating a lower rate of dissolution compared to SiO_2_. However, alumina faces its own challenges, including phase transformations in hydrothermal environments. It was reported that at 200 °C γ-Al_2_O_3_ undergoes a phase transition to hydrated boehmite in 10 h, [34,35] resulting in significant losses of surface area and Lewis acidic sites. This dual behavior of Al_2_O_3_—resistance to leaching yet susceptibility to structural changes—will be explored further in the following section.

### 2.1. Al_2_O_3_ as Support

#### 2.1.1. Al_2_O_3_ Supported Catalysts

Given the ample precedent of γ-Al_2_O_3_ as a support for steam reforming of methanol and ethanol, the use of γ-Al_2_O_3_ in aqueous-phase reforming (APR) of alcohols is equally prominently featured. γ-Al_2_O_3_-supported Pt and Ni have been tested as catalysts by various groups, often as catalysts to benchmark the performances of catalysts developed by these groups. The APR of methanol resulted in gaseous products that usually comprise CO_2_, CH_4_, CO, and H_2_; as such, it is common for researchers to report the specific hydrogen production rate. The performance of various Al_2_O_3_-supported Pt, Ni, and Cu catalysts in the APR of methanol is given in Table 2.

In one of the seminal works on the aqueous-phase reforming of oxygenated hydrocarbons, Cortright and Dumesic reported the use of Al_2_O_3_ nanofiber-supported Pt-based catalysts [36]. Methanol was amongst the oxygenated hydrocarbons tested. Impressively, they reported that the catalyst is stable on stream for at least a week with a 24-h-averaged specific hydrogen production rate of 40 mmol-H_2_/g-cat/h (Table 2, Entry 1). Li et al. demonstrated that Pt/Al_2_O_3_ is stable in terms of hydrogen production rate for 20 h on stream (Entry 2). The performance of Pt/Al_2_O_3_ in batch reactors is generally reported to be much higher than in flow reactors (Entry 3 and 4 vs. 1 and 2). However, Pt/Al_2_O_3_ was found to have exceptionally low activity when the APR of methanol was performed in the presence of NaOH (Entry 5) [37]. This could be due to the inherent instability of Al_2_O_3_ under strongly basic hydrothermal conditions [38].
molecules-29-04867-t002_Table 2Table 2Methanol APR for Al_2_O_3_-supported metal catalysts.Entry CatalystReaction ConditionH_2_ Prod. Rate (mmol/g-cat/h)Stability1 ^[a]^Pt/γ-Al_2_O_3_
^[b]^(3 wt.% Pt)Fixed-bed, 29 bar, 4.5 g catalyst,0.06 mL/min of 10 wt.% **methanol**,WHSV = 0.8 h^−1^40 (at 225 °C)Stable on stream for at least a week2 ^[c]^Pt/Al_2_O_3_
^[d]^(0.94 wt.% Pt)Fixed bed, 29 bar, 1 g catalyst, 0.05 mL/min of 10 wt.% **methanol**,WHSV = 3 h^−1^6 (at 210 °C)Stable for at least 20 h on stream3 ^[e]^Pt/Al_2_O_3_(2% wt.% Pt)Batch,20 bar,0.1 g catalyst,50 mL of 64 wt.% **methanol**,Time N.A.110 (at 240 °C)N/A4 ^[f]^Pt/Al_2_O_3_ (20 nm) ^[g]^(0.89% wt.%)Batch, 20 bar, 0.1 g catalyst, 15 mL of 37 wt.% **methanol**,1 h reaction86 (at 220° C)N/A5 ^[h]^Pt/Al_2_O_3_
^[i]^(0.2 wt.% Pt)Batch, 20 bar,0.1 g catalyst,15 g of 37 wt.% **methanol** in water and 0.3 g NaOH, 1 h reaction2.3 (at 220 °C)N/AWHSV is the weighted hour space velocity in g-feedstock/g-cat/h. N.D. = Not detected. ^[a]^ Data from Cortright and Dumesic. [36]. ^[b]^ Nanofiber from Argonide Corp. 500 m^2^/g. ^[c]^ Data from Li et al. [39]. ^[d]^ Pt dispersion from CO chemisorption = 70%. Pt particle size = 2 nm. CO chemisorption = 34 µmol/g. H_2_ chemisorption = 237 µmol/g. S_BET_ = 205 m^2^/g. ^[e]^ Data from Lin et al. [40]. ^[f]^ Data from Lv et al. [41]. ^[g]^ S_BET_ = 128 m^2^/g. Pore Vol. = 0.5 cm^3^/g, Pore Diameter = 15 nm. ^[h]^ Data from Liu et al. [37]. ^[i]^ S_BET_ = 159 m^2^/g.


Catalyst deactivation is ubiquitous in the APR of oxygenated hydrocarbons, especially for Ni- and Co-based catalysts, and it still poses significant challenges for achieving industrial application of APR. Wen et al. reported the stability of various transition metals supported on Al_2_O_3_ in the APR of glycerol by measuring the specific rate of hydrogen production on stream for up to 4 h (Figure 4) [42]. Pt/Al_2_O_3_ was found to be the most active and the most stable catalyst, followed by Cu/Al_2_O_3_ which shows an about 20% drop in activity over the 4 h on stream. The least stable catalysts are nickel and cobalt supported on Al_2_O_3_. Characterization of the spent catalysts revealed the loss of active sites as the cause for the loss of activity in Ni and Co catalysts. They attributed this to the crystallization of support, sintering of the metal particles, and carbon deposition, which occurs throughout the course of the reaction.

In another detailed study, Doukkali et al. examined spent γ-Al_2_O_3_ supported Ni, Pt, and Ni-Pt catalysts and revealed significant changes in the textural properties of the catalysts after the APR of glycerol [43]. The authors attributed the deactivation primarily to the formation of the boehmite phase (γ-AlOOH) from the support. This resulted in unfavorable changes in textural properties, such as surface area, porosity, and metal dispersion. The leaching of active metals was found to play an insignificant role in the deactivation process. However, the authors noted that catalysts synthesized via incipient wetness impregnation exhibited greater resistance to deactivation compared to those prepared by the sol–gel method, despite the latter showing higher initial glycerol conversion and gas conversion rates [44]. They ascribed this to the stability of the support due to its preparation method. Thus, modifying Al_2_O_3_ is an avenue explored by various research groups to prepare durable catalysts for the APR of alcohols.

This concept was exemplified in the work of Liu et al., who reported that incorporating ZnO into a series of Ni-xCu/Al_2_O_3_ catalysts (where x denotes the weight percentage of Cu) greatly improved the catalysts’ stability in methanol APR [45]. The bimetallic Ni and Cu catalysts demonstrated improved performance compared to both Ni/Al_2_O_3_ and Cu/Al_2_O_3_ (Table 3, Entry 1 and 2 vs. 3 and 4). Additionally, the methane production ratio was drastically reduced from 7.3% to 0.35% compared to Ni/Al_2_O_3_. As for stability, the control catalyst, Ni-8Cu/Al_2_O_3_ exhibited an activity loss of about 75% after a 72 h stability test, while that of ZnO-Ni-8Cu/Al_2_O_3_ was only 35% over the same duration (Entry 1 vs. 2). The presence of ZnO improves the catalyst stability by inhibiting the phase transformation of the support to the boehmite phase. X-ray photoelectron spectroscopy (XPS) analysis indicated that there were no significant changes in the oxidation states of the surface Ni and Cu particles and no carbon deposition on the spent catalysts’ surface.

Kalekar and Vaidya reported the APR of glycerol, sorbitol, and xylitol catalyzed by Ru/Al_2_O_3_ [46]. The catalyst has moderate activities of 2.5–3.5 mmol-H_2_/g-cat/h at 225 °C. The textural properties of the spent catalyst remained similar to that of the fresh catalysts (see footnotes e in Table 3). The specific surface area decreased to 195 m^2^/g, the pore volume decreased to 0.61 cm^3^/g, and the mean pore diameter decreased to 12 nm. However, the catalysts experienced an 18–24% loss in activity after 28 h time on stream.

#### 2.1.2. Catalysts Supported on Cobalt Aluminate and Hydrotalcite Related

γ-Al_2_O_3_ is known to undergo chemical weathering under hydrothermal conditions to form crystalline boehmite (γ-AlOOH), which can adversely affect its textural properties [47,48]. Inorganic dopants such as cations of Mg, Zr, and Ni have been shown to improve the resistance of modified alumina to chemical weathering [49]. In this section, we will discuss some examples of modified alumina as support for the aqueous-phase reforming of alcohols.

Modified aluminas, such as cobalt aluminate and nickel aluminate, are reported as supports or catalysts for the aqueous-phase reforming (APR) of methanol and glycerol. Cobalt aluminate (Co[x]Al[y], where x and y indicate the relative amount of Co or Al) is notable for its hydrophobic nature, low surface acidity, and exceptional thermal, mechanical, and pH stability [50], making it a logical choice as a support for catalysts in the APR of alcohols. Additionally, spinel mixed-metal oxides like NiAl_2_O_4_ are recognized for enhancing the stability of catalysts under hydrothermal conditions [51,52].

Reynoso et al. have conducted extensive studies on using cobalt aluminate as catalysts or as a support for Pt in the aqueous-phase reforming (APR) of glycerol. While cobalt aluminate reduced at 600 °C (0.625CoAl-600) is active in the APR of glycerol, its stability under these conditions is found to be insufficient, with a 48% reduction in the hydrogen production rate after 30 h on stream (Table 4, Entry 1) [53]. However, Pt supported on 0.625CoAl, prepared by wet impregnation, remains stable on stream for 100 h (Entry 2) [54]. It is important to note that this stability test was conducted under a very low flow rate (WHSV = 0.68 h^−1^), thereby resulting in a very low hydrogen production rate. In a separate study, they also examined the effects of various process parameters on glycerol APR using 0.3 wt.% Pt/0.625CoAl [55]. They found that while decreasing contact time by increasing WHSV resulted in a higher H_2_ production rate (Entry 3), the overall conversion of glycerol to gases decreased.

Deactivation of cobalt aluminate during the APR of glycerol is investigated by extensive characterization of the spent catalyst [53]. Reynoso et al. reported that the specific surface area and pore volume of the spent catalysts notably increased after use, while the average pore size decreased. This increase in specific surface area was more pronounced in samples with a lower Co/Al ratio, suggesting the involvement of aluminum-based compounds. X-ray diffraction (XRD) analysis showed that FCC metallic cobalt was present in all spent catalysts, and sintering increased the cobalt crystallite size, particularly in Co_3_O_4_ samples. H_2_-TPR analysis indicated the re-oxidation of cobalt during the APR process, which contributed to the catalyst deactivation. The formation of CoO and gibbsite was also observed, indicating strong oxidizing conditions. Additionally, Raman spectroscopy revealed the presence of both defect/amorphous and graphitic carbon on the catalyst surfaces, with a higher proportion of graphitic carbon, which is associated with deactivation.

Lv et al. reported the aqueous-phase reforming (APR) of methanol using cobalt-aluminate-supported Pt catalysts [41]. They emphasized the development of a series of Pt/Co_xAl (x = Co/Al ratio) catalysts derived from calcined layered double hydroxides. The study found that the catalysts’ performance strongly depends on the Co/Al ratio and the calcination temperature. Notably, the Pt/Co_2Al catalyst calcined at 700 °C (Co_2Al-c700) exhibited exceptional activity and low CO selectivity (Table 4, Entry 4). The interactions between Pt and the support, coupled with abundant oxygen vacancies, were proposed to enhance catalytic performance. Furthermore, they revealed that both Pt and metallic Co are active in methanol decomposition, while water activation on the support facilitates the conversion of intermediate formate species into CO_2_ and H_2_. The stability of the optimal catalyst was evaluated through ten recycling rounds, during which they observed a decrease in the hydrogen production rate of about 9%, but negligible changes in CO selectivity. Through detailed XPS analysis, they found that the amounts of surface Co^0^ and Co^2+^ increased after ten cycles, while the amount of Co^3+^ on the surface decreased from 27.7% to 19.3%. In addition, an increase in adsorbed oxygen species and the possibility of oxygen vacancies were observed. The leaching of Pt was not observed, while ppm levels of Co and Al were found in the liquid after ten cycles.

#### 2.1.3. Nickel Aluminate

Morales-Marín et al. reported the use of nickel aluminates as catalysts for the APR of glycerol [56]. They found that catalysts synthesized by reducing the calcined nickel aluminate at 700 °C or 850 °C gave the best performance. Glycerol conversion and NiAl_2_O_4_ reduction at 850 °C were found to decrease by 47% after 50 h on stream. Hydrogen yield, however, remained more stable throughout the 50 h on stream. The losses in specific hydrogen production rates were approximately 12%, respectively (Table 5, Entry 1). Characterization of the spent catalyst revealed that with increasing time on stream (TOS), the Ni particle sizes increased (fresh: 11.6 nm, 2 h: 41.5 nm, 50 h: 44.1 nm). Similarly, the leaching of Ni increased with increasing TOS (2 h: 0.19%, 50 h: 5.4%). They observed a drastic decrease in exposed Ni after 2 h TOS of from 3.47 m^2^/g in the fresh catalyst to 0.23 m^2^/g in the spent catalyst. They postulated that this is due to the formation of core–shell particles under APR conditions [43].

Li et al. reported the application of Pt on NiAl_2_O_4_ in the APR of methanol (Table 5, Entry 2) [39]. The catalyst was found to lose about 10% of methanol conversion to gas after 600 h time on stream. This work probably represents the longest on-stream test reported in APR thus far. We note that information on the gas selectivity was absent, thus it is difficult to assess if hydrogen selectivity remained the same after 600 h on stream. Characterization of the spent catalysts by XRD revealed the formation of NiO, and TGA analysis showed an additional 3% by weight in materials on the catalysts that could be lost when the catalysts were heated.

#### 2.1.4. SiO_2_-Al_2_O_3_

SiO_2_ as a support has limited hydrothermal stability due to its dissolution at elevated temperatures and changes in textural properties [35,57]. However, it was found that the addition of aluminium oxide improves the hydrothermal stability of the silica-based material. This finding is reflected in the APR reaction discussed below.

Dumesic and co-workers reported that 0.75 wt.% Pt/SiO_2_ lost more than 20% of its activity in the APR of ethylene glycol over 24 h [33]. However, they noted that 0.79 wt.% Pt/SiO_2_-Al_2_O_3_ did not experience the same issue. In their report, 0.79 wt.% Pt/SiO_2_-Al_2_O_3_ has a much higher H_2_ turnover frequency (TOF) of 4.6 min^−1^ compared to 0.75 wt.% Pt/SiO_2_, where it was 0.7 min^−1^; although this is lower than with Pt/Al_2_O_3_ (H_2_ TOF of 7 min^−1^) under the same reaction conditions.

Wen et al. reported that a 5.1 wt.% Pt/HUSY (SiO_2_/Al_2_O_3_ = 4.8) catalyst showed no observable deactivation in the aqueous-phase reforming (APR) of glycerol for about 4 h on stream (Figure 5) [42]. The average hydrogen production rate over this 4 h period is 19.7 ± 1.7 mmol-H_2_/g-cat/h. Additionally, Pt/HUSY maintained a higher metal surface area of 13.3 m^2^/g for the spent catalyst compared to 0.4 m^2^/g for Pt/SiO_2_ and 7.2 m^2^/g for Pt/Al_2_O_3_ after glycerol APR.

The use of commercially available 65 wt.% Ni/SiO_2_-Al_2_O_3_ in the APR of glycerol from biodiesel waste is reported by Seretis and Tsiakaras [59]. However, no information on stability was provided.

### 2.2. CeO_2_ as Support

#### 2.2.1. CeO_2-_Supported Catalysts

CeO_2_ possesses oxygen vacancies that can be tuned via the preparation method. These oxygen vacancies can enhance the dispersion of noble metals through strong metal-support interactions and facilitate the water–gas shift reaction, which is responsible for generating hydrogen from the reaction of water with the CO formed in APR. However, CeO_2_-supported Ni- or Pt-based catalysts are generally not reported to have high stability under hydrothermal conditions. Early work by Dumesic and co-workers on the APR of ethylene glycol highlighted the potential for the dissolution of CeO_2_ over extended periods under APR conditions. They found that Pt/CeO_2_ not only exhibited low H_2_ turnover frequency but it also experienced significant deactivation with a more than 20% loss in activity over 24 h [33]. Ciftci et al. reported similar findings for the APR of glycerol with 2.7 wt.%Pt/CeO_2_ [60].

Wu et al. reported the APR of glycerol, which is directly derived from biodiesel byproducts, using mesoporous Ni-Cu/CeO_2_ catalysts for hydrogen production [61]. The inclusion of copper in the catalysts enhances the water–gas shift reaction and suppresses methane formation, thereby improving hydrogen yield by reducing the amount of CO formed from 6.1% to 2.7%, and methane from 1.4% to 0.12% (Table 6, Entry 1 and 2). Additionally, the study suggests that adding CaO helps to adsorb the CO_2_ formed, further reducing the amount of CO formed to 0.61% (Entry 3). The research also examines the impact of reaction temperature and CaO addition on APR performance, revealing that higher temperatures increase hydrogen production but do not affect selectivity. The catalysts demonstrated stable performance over 50 cycles, showing only about a 14% reduction in their initial hydrogen production rate. The CO_2_ and CO contents of the gaseous product increase with the loss in H_2_ production rate. This stability underscores their potential for sustainable hydrogen production from biodiesel byproducts.

Lu et al. reported three ceria catalysts prepared via a photochemical reduction method [62]. They suggested that this method prevents the aggregation and migration of active metal particles that typically occur during thermal reduction at high temperatures, thus ensuring a high dispersion of the active metals. The lanthanum-modified catalyst—PtLa/CeO_2—_exhibited both a higher initial hydrogen production rate and greater stability compared to Pt/CeO_2_-HT, which is hydrothermally prepared CeO_2_. Notably, the latter demonstrated an 87% reduction in hydrogen production rate after five cycles (Table 7, Entry 1 vs. 2). The authors observed that Pt/CeO_2_-HT underwent unfavorable morphological changes leading to significant leaching of Pt, a process they attributed to catalyst deactivation during the recycling experiments. They credited the stability of PtLa/CeO_2_ to lanthanum’s role in reducing Pt leaching through stronger metal-support interactions, and to the increase in oxygen vacancies that accelerate CO removal from the CeO_2_ surface. Without these modifications, CO would likely lead to the formation of carbonates of cerium, compromising the structural integrity of the support.

#### 2.2.2. CeO_2_-Supported Catalysts for Low-Temperature Aqueous Reforming (<150 °C)

Operating the aqueous-phase reforming (APR) of alcohols at the same temperature (80–90 °C) as Proton Exchange Membrane (PEM) fuel cells is advantageous because it minimizes energy losses from heating or cooling hydrogen to match the fuel cell’s temperature [63].

Homogeneous catalysts, such as the Ruthenium complex reported by Beller and co-workers, are highly active and can produce hydrogen at a turnover frequency of 2670 h⁻¹ at 90 °C in the presence of a strong base or caustic aqueous phase reforming [64]. Pt supported on a tailor-made CeO_2_ support has demonstrated similar efficiency in proof-of-concept studies. However, heterogeneous catalysts offer the advantage of easier separation from reaction mixtures compared to homogeneous catalysts, making them more practical for industrial applications. In this section, we will focus on studies that utilized CeO_2_ as a support to demonstrate the feasibility of the low-temperature APR of methanol with heterogeneous catalysts.

Zhang et al. developed a catalyst for the APR of methanol, employing a novel approach by combining Pt single-atoms with frustrated Lewis pairs (FLPs) on a porous nanorod CeO_2_ support (Pt_1_/PN-CeO_2_) [65]. This dual-active-site strategy enables efficient hydrogen production from methanol at a low temperature of 135 °C while significantly reducing the formation of carbon monoxide by-products. Despite the Pt_1_/PN-CeO_2_ catalyst achieving only a modest hydrogen production rate of 3.7 mmol-H_2_/g-cat/h at 135 °C (Table 8, Entry 1), it showed a significant improvement over Pt on Al_2_O_3_, TiO_2_, and carbon supports, which produced hydrogen at a negligible rate (0.02–0.1 mmol-H_2_/g-cat/h) at the same temperature. However, despite the low reaction temperature, the catalyst experienced a 20% reduction in hydrogen production rate after 10 one-hour cycles. However, the CO levels were consistently maintained below 0.03%. This decline in activity was attributed by the authors to the mobility of peripheral Pt atoms, which led to the sintering of the Pt active sites.

In contrast, Chen et al. reported on a Pt_1_/PN-CeO_2_ catalyst [66], synthesized using a modified ascorbic acid-assisted reduction route [67]. This catalyst demonstrated stability (measured by methanol conversion) for at least 110 h on stream at 300 °C, despite an initial decrease in activity attributed to the leaching of Pt. However, since the reaction was conducted at an initial pressure of 1 atm, the reactants’ phase deviates from that typically observed in aqueous-phase reforming (APR) and is therefore not included in our tabulation.

Guo et al. present a significant advancement in hydrogen generation from the aqueous-phase reforming of methanol with KOH at low temperatures [68]. By increasing the loading of Pt nanoparticles from 0.36 wt.% to 1 wt.% and utilizing porous nanorods of CeO_2_ with abundant oxygen vacancies as support (Pt/PN-CeO_2_), the study achieves efficient hydrogen production at 90 °C, with an impressive rate of 73.4 mmol-H_2_/g-cat/h (Table 8, Entry 2). The presence of oxygen vacancies not only enhances the electronic density of the supported Pt nanoparticles, facilitating methanol activation, but also promotes water activation. The efficacy of this approach was further demonstrated by performing the APR of methanol at 60 °C, where hydrogen production continued at a rate of 2.8 mmol-H_2_/g-cat/h. Remarkably, at this low temperature, no CO formation was detected. Despite the reduced reaction temperature of 60 °C, a 22% decrease in the specific hydrogen production rate was observed.
molecules-29-04867-t008_Table 8Table 8APR of methanol at low temperatures enabled by Pt/CeO_2_.EntryCatalystReaction ConditionH_2_ Prod. Rate (mmol/g-cat/h)Stability1 ^[a]^Pt/PN-CeO_2_
^[b]^(0.36 wt.%Pt)Batch,40 bar,0.05 g catalyst, 58 mL of 63.8 wt.% **methanol** in water1 h reaction20.4 (165 °C)3.7 (at 135 °C)20% loss (to 16) after 10 cycles of one hour each at 165 °C2 ^[c]^Pt/PN-CeO_2_
^[d]^(1 wt.% Pt)Batch,1 atm0.005 g catalyst,5 mL of 56.4 wt.% **methanol** in 8M KOH (aq)1 h reaction2.8 (at 60 °C)73.4 (at 90 °C)22% loss (to 2.1) after 10 cycles of one hour each at 60 °C^[a]^ Data from Zhang et al. [65]. ^[b]^ PN-CeO_2_: porous nanorod CeO_2_. Pt particle size (Transmission Electron Microscopy) = 1.3 ± 0.3 nm. S_BET_ = 122 m^2^/g. ^[c]^ Data from Guo et al. [68]. ^[d]^ Pt particle size (Transmission Electron Microscopy) = 1.4 ± 0.1 nm.


### 2.3. ZrO_2_ as Support

#### 2.3.1. ZrO_2_-Supported Catalysts

ZrO_2_ has attracted interest as a support for heterogeneous catalysts due to its unique blend of surface acidity, surface basicity, high thermal stability, and resistance to reduction [69,70]. Goplan reported that ZrO_2_ membranes showed the most resistance to hydrothermally induced sintering among the ceramics tested [71]. However, tetragonal and monoclinic ZrO_2_ were both reported to lose surface area when subjected to hydrothermal reaction at 250 °C for 10 h [72]. This low specific surface area could affect the performance of a heterogeneous catalyst [73].

Among the catalysts studied in the APR of methanol, Strekrova et al. reported that 9.4 wt.% Ni/ZrO_2_ demonstrated a respectable 60 mmol-H_2_/g-cat/h under continuous flow operation (Table 9, Entry 1) [74]. They reported an 18% loss in specific hydrogen production rate, together with an increase in CO selectivity, after 12 h TOS. Minimal changes in textural properties were observed in the spent catalysts, but there was significant sintering in the spent catalyst—the particle sizes increased from 12.7 nm for NiO of the fresh catalyst to 47.6 nm for Ni of the spent catalysts. They explored various mixed oxides of La, Ce, and Zr to further improve the activity and stability of the approximately 10 wt.% Ni-based catalysts, which will be discussed in the next section.

Contreras et al. studied three catalysts with Mo_2_C supported on ZrO_2_ for the APR of ethanol [75]. They found that the optimal catalyst was that with β-Mo_2_C supported on monoclinic ZrO_2_ or *m*-ZrO_2_. The other two catalysts, β-Mo_2_C/*t*-ZrO_2_ and α-MoC/*m*-ZrO_2_, displayed very low selectivity towards H_2_. Nevertheless, their optimal catalyst lost 41% of its initial activity after four cycles of ethanol APR (Table 9, Entry 2). XPS revealed no significant changes in textual properties between the fresh and spent catalysts. However, the distribution of Mo/Zr and Mo/C exhibited significant changes. They attributed the deactivation to the disappearance of oxycarbide species on the catalyst’s surface after the reaction as observed from the XPS spectrum of the spent catalysts.
molecules-29-04867-t009_Table 9Table 9ZrO_2_-supported catalysts for APR of alcohols.EntryCatalystReaction ConditionH_2_ prod. Rate (mmol/g-cat/h)Stability1 ^[a]^Ni/ZrO_2_
^[b]^(9.4 wt.% Ni)Fixed bed,32 bar,1.5 g catalyst,2 mL/min of 5 wt.% **methanol** in water.WHSV = 80 h^−1^60 (at 230 °C)18% loss in hydrogen production rate (to 49) after 12 h TOS.CO/CH_4_ selectivity changes to 7.8%/1.1% from 4.7%/1.5%2 ^[c]^β -Mo _2_ C/*m*-ZrO_2_
^[d]^(10 wt.% Mo)Batch,6 bar,0.04 g catalyst,15 mL of 0.4M **ethanol**1.5 h reaction20 (at 250 °C) ^[e]^41% loss (to 12) after 4 cycles of 1.5 h each.WHSV is the weighted hour space velocity in g-feedstock/g-cat/h. ^[a]^ Data from Strekrova et al. [74]. ^[b]^ Ni particle size = 26 nm. Ni dispersion from H_2_ chemisorption = 3.9%. Ni surface area from H_2_ chemisorption = 25.9 m^2^/g. S_BET_ = 60 m^2^/g. Pore Vol. = 0.26 cm^3^/g. Pore Diameter = 11.7 nm. ^[c]^ Data from Pavesi Contreras et al. [75]. ^[d]^ S_BET_ = 39 m^2^/g. Pore Vol. = 0.15 cm^3^/g. Pore Diameter = 11 nm. ^[e]^ H_2_ production rate is taken from the 2nd cycle as there is a large increase in H_2_ selectivity after the first cycle.


#### 2.3.2. Mixed Oxides of ZrO_2_

Given the potential hydrothermal instability of CeO_2_ as a support due to its deactivation via the formation of CeCo_3_OH [76], various groups have tried to rectify this by combining various oxides with the goal of obtaining a more stable aqueous-phase reforming catalyst.

Larimi et al. reported on the APR of glycerol using Pt_0.05_Ce_x_Zr_0.95−x_O_2_ ternary solid solution catalysts, with a focus on the influence of the cerium/zirconium ratio on catalyst performance [77]. The Pt_0.05_Ce_0.475_Zr_0.475_O_2_ catalyst achieved optimal performance, showing the highest glycerol conversion (99.8%), carbon-to-gas conversion (95%), hydrogen yield (93%), and selectivity for hydrogen (98%). This enhanced performance is attributed to factors such as the Pt oxidation state, active metal dispersion, surface area, and particle size, all of which are governed by the Ce-to-Zr ratio. Significantly, this catalyst also demonstrated remarkable stability, maintaining a high activity rate of 91 mmol-H_2_/g-cat/h for at least 50 h of continuous operation without deactivation (Table 10, Entry 1). In addition, Pt_0.05_Ce_0.475_Zr_0.475_O_2_ was found to demonstrate the same stability profile at various weighted hour space velocities (0.12 h^−1^ to 3.6 h^−1^). The spent catalysts did not manifest any sintering of Pt nanoparticles as they maintained a size of about 5.3 nm in both the fresh and spent catalysts. Detailed XPS analysis suggested that incorporating Pt into the ceria-zirconia matrix prevents Pt segregation under APR conditions.

Strekrov et al. reported on the APR of methanol using nickel-supported catalysts on mixed-oxide supports, specifically zirconium, cerium, and lanthanum oxides, in terms of hydrogen production efficiency and stability [74]. Their study revealed that nickel catalysts supported on combinations of cerium and zirconium oxides exhibited superior performance compared to those supported solely on CeO_2_ or ZrO_2_. Among the tested catalysts, the 9.3 wt.% Ni/25Ce-Zr catalyst was the most active, achieving a hydrogen production rate of 151 mmol/g-cat/h (Table 10, Entry 2). Further improvements in stability were observed when lanthanum was incorporated into the support (Table 10, Entry 3). The most stable catalyst, containing 9 wt.% Ni/10La-Zr, experienced only an 8% loss in hydrogen production rate after 12 h of time on stream (TOS), recording a rate of 118 mmol-H_2_/g-cat/h (Table 10, Entry 4). The authors attributed this activity loss primarily to the sintering of Ni particles. It is important to note, however, that the comparison was made between NiO particles in the fresh catalysts and Ni in the spent catalysts. The support metal particles remained virtually unchanged after 12 h TOS. The leaching of Ni for Ni/25Ce-ZrO_2_ and Ni/10La-ZrO_2_ was almost negligible, while Ni/17Ce-5La-ZrO_2_ lost about 0.8% of its Ni content after 12 h on stream.
molecules-29-04867-t010_Table 10Table 10Mixed oxides of ZrO_2_ supporting Pt or Ni catalysts in APR of methanol.EntryCatalystReaction ConditionH_2_ Prod. Rate (mmol/g-cat/h)Stability1 ^[a]^Pt_0.05_Ce_0.475_Zr_0.475_O_2_
^[b]^Fixed-bed,50 bar,0.25 g catalyst,0.61 mL/min of10 wt.% **glycerol**WHSV = 2.45 h^−1^91 (at 250 °C)Virtually no loss after 50 h on stream2 ^[c]^Ni/25Ce-ZrO_2_
^[d]^(9.3 wt.% Ni)Fixed bed,32 bar,1.5 g catalyst,2 mL/min of 5 wt.% **methanol** in water.WHSV = 80 h^−1^151 (at 230 °C)30% loss in hydrogen production rate (to 106) after 12 h TOS.CO/CH_4_ selectivity changes to 7.8%/1.1% from 4.7%/1.5%3 ^[c]^Ni/17Ce-5La-ZrO_2_
^[e]^ (10.1 wt.% Ni)128 (at 230 °C)15% loss in hydrogen production rate (to 109) after 12 h TOS.CO/CH_4_ selectivity changes to 5.9%/3.3%% from 4.6%/2.1%4 ^[c]^Ni/10La-ZrO_2_
^[f]^ (9.0 wt.% Ni)129 (at 230 °C)8% loss in hydrogen production rate (to 118) after 12 h TOS.CO/CH_4_ selectivity changes to 5.2%/2.6% from 4.2%/3.0%WHSV is the weighted hour space velocity in g-feedstock/g-cat/h. ^[a]^ Data from Larimi et al. [77]. ^[b]^ Pt dispersion = 39%. Pt surface area = 5 m^2^/g. CO chemisorption = 101 µmol/g. H_2_ chemisorption = 1700 µmol/g. S_BET_ = 81 m^2^/g. Pore Vol. = 0.13 cm^3^/g. Pore Diameter = 6 nm. ^[c]^ Data from Strekrov et al. [74]. ^[d]^ Ni particle size = 27.3 nm. Ni dispersion from H_2_ chemisorption = 3.7%. Ni surface area from H_2_ chemisorption = 24.7 m^2^/g. S_BET_ = 83 m^2^/g. Pore Vol. = 0.26 cm^3^/g. Pore Diameter = 10.9 nm. ^[e]^ Ni particle size = 35.6 nm. Ni dispersion from H_2_ chemisorption = 2.9%. Ni surface area from H_2_ chemisorption = 19.1 m^2^/g. S_BET_ = 114 m^2^/g. Pore Vol. = 0.36 cm^3^/g. Pore Diameter = 3.7 nm. ^[f]^ Ni particle size = 21.9 nm. Ni dispersion from H_2_ chemisorption = 4.6%. Ni surface area from H_2_ chemisorption = 30.7 m^2^/g. S_BET_ = 69 m^2^/g. Pore Vol. = 0.25 cm^3^/g. Pore Diameter = 12.6 nm.


Bastan et al. studied the aqueous-phase reforming (APR) of glycerol, with a focus on producing alkanes using various 10 wt.% Ni/Ce_X_Zr_1-x_O_2_ catalysts synthesized by coprecipitation [78]. In contrast to the nickel catalyst supported on Ce_0.25_-Zr0._75_O_2_, which was synthesized by incipient wetness impregnation of the support, their 10 wt.% Ni/Ce_0.3_Zr_0.7_O_2_ catalysts maintained alkane selectivity over six cycles of 25 h each. However, since the impact of time on stream on hydrogen production rate was not discussed—given it was not the focus of their work—a direct comparison of stability is not feasible.

### 2.4. Other Metal-Supported Catalysts

#### 2.4.1. TiO_2_-Related

TiO_2_ is the only common oxide that is predicted to be hydrothermally stable over a large pH window at 200 °C [21]. However, reports on its application as a support in the aqueous-phase reforming (APR) of alcohols are relevantly scant.

In the APR of ethylene glycol reported by Shabaker et al., Pt/TiO_2_ was found to have superior turnover frequency compared to common supports such as SiO_2_, ZrO_2_, SiO_2_-Al_2_O_3_, Al_2_O_3_, ZnO, and carbon [33]. They commented that no significant loss in activity was observed after 24 h on stream. However, they noted the potential for TiO_2_ to degrade via sintering and undergo phase transformation, resulting in a loss in surface area. We note that the work of Lin et al. was cited [71]. This work explored the hydrothermal stability of TiO_2_ at temperatures beyond 450 °C and at atmospheric pressure, thus it might have limited relevance to the APR of alcohols. In the work of Ellilott et al., rutile phase TiO_2_ was shown to demonstrate good hydrothermal stability and low coke formation [79].

Detailed studies on the stability of TiO_2_ as a support for the aqueous-phase reforming (APR) of alcohols are scarce. Lin et al. reported the APR of methanol catalyzed by Pt/TiO_2_ and Ni/TiO_2_ at 190 °C and 240 °C, respectively [40,80]. Pt/TiO_2_ exhibited a modest specific hydrogen production rate of 15 mmol-H_2_/g-cat/h and produced a very low amount of CO (0.02 mol% of the H_2_ produced) [80]. In contrast, Ni/TiO_2_ achieved a decent specific hydrogen production rate of 21 mmol-H_2_/g-cat/h but suffered from high selectivity to CO (16.6 mol% of the H_2_ produced) [40]. Since these catalysts were not the primary focus of their studies, the authors did not explore their stability.

Nozawa et al. investigated various noble metals supported on P25 TiO_2_ in the APR of ethanol, with selected results compiled in Figure 6 [81,82]. For monometallic catalysts, Rh/TiO_2_ was found to be the most active catalyst in the APR of ethanol, although it produced a large amount of CH_4_ (33 mol. %). The activity of Rh/TiO_2_ could be further improved by adding an equimolar amount of Re, but the amount of CH_4_ in the gaseous product remained high (32 mol. %). Ir-Re/TiO_2_, while slightly less active in terms of hydrogen production rate, produced significantly less CH_4_ (5.8 mol. %). However, as these results were obtained in a batch reactor without recycling the spent catalysts, extracting reliable stability information is challenging.

Zhao et al. reported Pt and Ru supported on TiO_2_ as catalysts for the APR of ethanol [83]. The APR reactions were performed in a batch reactor. They found that at a low Ru loading of 0.5 wt.% the selectivity of ethanol APR towards CH_4_ decreases significantly. The optimal catalyst was reported to be 1 wt.% Ru–2 wt.% Pt/TiO_2_ with a hydrogen production rate of 294 mmol-H_2_/g-cat/h for the freshly reduced catalyst. The purity of hydrogen is about 65% with 13.4 mol.% CO_2_ and 17.7 mol.% CH_4_. CO was not detected by the authors. The stability in terms of hydrogen production rate cannot be extracted, but there is a notable decrease in ethanol conversion for one of the reported catalysts after 1 h.

Cifti et al. reported bimetallic Pt and Re supported on TiO_2_ in the APR of glycerol [60]. The bimetallic Pt-Re/TiO_2_ has a turnover frequency of 345 h^−1^ which is 2.3 times that of Pt/TiO_2_. However, Pt-Re/TiO_2_ seems to be less selective towards H_2_. They observed a decrease in the H_2_-to-CO_2_ ratio over the course of 380 min.

#### 2.4.2. MgO Hydrotalcite and Related Support

Hydrotalcites or double-layered hydroxides generally consist of a divalent cation (Mg^2+^, Mn^2+^, Fe^2+^, Cu^2+^, Zn^2+^, Ca^2+^), and a trivalent cation (Al^3+^, Fe^3+^, Ga^3+^) balanced by anions such as chloride or carbonate [84]. These have been extensively studied in steam reforming to capture CO_2_ reversibly, or sorption-enhanced steam reforming (SESR) [8,10]. In terms of the steam reforming of ethanol and glycerol, the group of Vaidya reported the use of a hydrotalcite-related support to achieve low selectivity for CO [85,86,87].

Huang et al. reported the aqueous-phase reforming (APR) of methanol over La-promoted Ni-based hydrotalcite catalysts to produce hydrogen [88]. A La-promoted hydrotalcite catalyst, 5La-NiMgAl, exhibited a superior APR reactivity compared to an unpromoted one (0La-NiMgAl). The 5La-NiMgAl catalyst exhibited remarkable stability, retaining 80% of its activity, while 0La-NiMgAl retained only 28% over the same period, as shown in Table 11 (Entry 1 vs. 2).

The authors attributed this increased stability to two main factors. First, there was a 12% reduction in nickel leaching in 5La-NiMgAl compared to La-NiMgAl, resulting in more catalytically active metal remaining in the catalyst. Second, there was an increase in the concentration of medium-strength basic sites (refer to Table 11, Footnotes b and c), due to the introduction of La. This is critical in maintaining the structural integrity of the catalyst by forming LaCO_3_OH, which adsorbs CO_2_ and prevents the structural breakdown of the catalyst during the reaction process.

Manfro et al. synthesized a series of Ni-Cu hydrotalcite-like catalysts for the APR of glycerol [89]. They found that Ni5Cu (Table 11, Entry 4) demonstrated an enhanced rate of hydrogen production compared to the NiMg catalyst (Entry 3). Specifically, Ni5Cu achieved a hydrogen production rate of 10 mmol-H_2_/g-cat/h, although this rate is considered modest given that the reaction was performed at 250 °C (Entry 3 vs. 4). The introduction of Cu resulted in a drastic decrease in Ni particle size sizes (refer to Table 11, Footnotes e and f). However, the authors revealed an increase in the size of nickel particles in both the NiMg and Ni5CuMg catalysts after 6 h of time on stream (TOS) compared to their freshly reduced counterparts, but no deactivation of the catalysts was observed after 6 h of TOS.

Cruz et al. reported the APR of ethanol using nickel catalysts derived from hydrotalcite precursors (Ni, Mg, and Al) [90]. The study highlights the preparation of these catalysts with varying nickel content and their substantial activities and selectivities in hydrogen production. The optimal catalyst 20NiHTC consists of 16.5 wt.% NiO, 51.3 wt.% MgO, and 21 wt.% Al_2_O_3_. This catalyst achieved an ethanol conversion of 70%, a hydrogen selectivity of 78%, and produced less than 1% CO at 230 °C. The authors reported a BET specific surface area of 286 m^2^/g, pore volume of 0.694 cm^3^/g, and Ni particle size of 8 nm for this catalyst. However, we did not include this in Table 11 due to the difficulty in obtaining data on hydrogen production rate and stability.

### 2.5. Carbon-Supported Catalysts

It has been reported by several groups that common supports such as SiO_2_ and CeO_2_ potentially dissolve or degrade under hypothermal conditions [91]. Dusmesic suggested that due to the hydrophobic nature of carbon, it might demonstrate hydrothermal stability. This point is echoed by Wang and co-workers, who investigated the APR of glycerol using Pt-Re/AC catalysts [92,93,94].

#### 2.5.1. Activated Carbon

Pt supported on activated carbon (AC) generally has low activity in the APR of alcohols. However, Shabaker et al. reported that Pt supported on Norit^®^ SX 1G exhibited similar performance to Pt/TiO_2_ and Pt/Al_2_O_3_ in the APR of ethylene glycol [33].

Various Pt/AC catalysts have demonstrated stability for 4–25 h on stream, as reported by different research groups. For example, Wen et al. reported that a 4.2% Pt/AC catalyst achieved a hydrogen production rate of 18 mmol-H_2_/g-cat/h during a 4 h run at 230 °C in the APR of glycerol (Table 12, Entry 1) [42]. Additionally, Kim’s group found that a 7 wt.% Pt/AC catalyst remained stable for 25 h on stream and exhibited an activity of 38 mmol-H_2_/g-cat/h at 250 °C for the APR of ethylene glycol (Table 12, Entry 2) [95]. Furthermore, they reported that bimetallic catalysts based on Pt and Fe enhanced the activity of activated-carbon-supported catalysts in the APR of ethylene glycol (Table 12, Entry 3) [96]. Specifically, the Pt-Fe/AC catalyst maintained a hydrogen production rate of 97 mmol-H_2_/g-cat/h for 90 h on stream at 250 °C.

Some notable works on enhancing the activity of activated carbon support catalysts are given below. As the authors did not present data on stability, we will not include them in Table 12.

In addition to bimetallic Pt and Fe, Wang and co-workers reported the synergy between Pt and Re in the APR of glycerol at 225 °C [92,93,94]. The H_2_ turnover frequency in 3 wt.% Pt and 3 wt.% Re supported on AC was about 12 times that of only 3 wt.% Pt supported on the same AC. However, the hydrogen selectivity decreased with increasing rhenium content. Cifti et al. reported similar findings in the APR of glycerol [60].

Wang et al. reported the use of KMnO_4_ treatment on AC to increase its hydrophilicity. The modified PtMnK/AC catalyst exhibited an increase in hydrogen yield in the APR of methanol compared to Pt/AC, with the former producing a H_2_ yield of 51% while the latter producing a H_2_ yield of 23% [97].

Cu supported on activated carbon was reported to have a very low activity in the APR of methanol of about 9.7 mmol-H_2_/g-cat/h [98]. It was found to suffer from severe sintering of Cu particles, with particle sizes increasing from 6.4 nm in the freshly reduced catalyst to 127 nm after being subjected to 1.25 h of methanol APR [98].

#### 2.5.2. Ordered Mesoporous Carbon Support

The high specific surface area and hydrothermal stability of carbon make it an attractive choice as a support for the aqueous-phase reforming (APR) of alcohols. However, concerns have been raised about unfavorable textural properties, such as irregular pore distribution, in activated carbon (AC), which can hinder the mass transfer of alcohols during the APR reaction [95]. To address these issues, Kim’s group has extensively reported on the use of ordered mesoporous carbon, which will be discussed below.

Earlier work by the group of Kim disclosed various Pt catalysts supported on mesoporous carbon synthesized via a hard template. Their earliest work disclosed the use of rod-like ordered mesoporous carbon (CMK-3) as support for Pt in the APR of ethylene glycol [95]. They varied the loading of Pt from 1 to 10 wt.% and found that 7 wt.% Pt loading is the optimal. A catalyst consisting of 7 wt.% Pt/CMK-3 was able to produce hydrogen at 91 mmol/g-cat/h for 25 h on stream (Table 13, Entry 1) without deactivation being observed. The low-angle powder XRD peaks for the catalyst before and after the reaction suggested that the structural integrity of the ordered mesoporous carbon was maintained. However, a slight degradation in the micropore structures of CMK-3 was deduced from the reduction in surface area and pore volume after the reaction.

By varying the mesoporous silica hard templates from SBA-15 to KIT-6, a related hollow-type framework ordered mesoporous carbon CMK 9 can be synthesized. CMK-9 was studied in a separate work by the same group [99]. The catalyst 7 wt.% Pt/CMK-9 demonstrated the highest hydrogen production rate at 152 mmol-H_2_/g-cat/h for 25 h on stream (Table 13, Entry 2). Negligible aggregation of Pt particles was observed after the reaction, as opposed to Pt/CMK-3, which experienced an increase in particle size from 1.8 nm to 7 nm. The 3D structure of CMK9 is proposed to improve the stability by providing a larger surface area (S_BET_ of CMK = 1717 m^2^/g vs. S_BET_ of CMK-3 = 770 m^2^/g) to achieve better dispersion and prevent particle aggregation.

In a separate report by the Kim group, they reported that bimetallic Pt and Fe supported on CMK-9 improved the hydrogen production rate by 44% in the APR of ethylene glycol (Table 13, Entry 3 vs. 4) [96]. The catalyst was found to be stable on stream for 90 h.
molecules-29-04867-t013_Table 13Table 13Ordered-mesoporous-carbon-supported catalysts for the APR of ethylene glycol.EntryCatalystReaction ConditionH_2_ Prod. Rate (mmol/g-cat/h)Stability1 ^[a]^7 wt.% Pt/CMK-3 ^[b]^Fixed bed,45 atm,0.3 g catalyst,10 wt.% **ethylene glycol** in water,0.1 mL/min,WHSV = 2 h^−1^91 ^[c]^ (at 250 °C)No deactivation was observed after 25 h TOS2 ^[d]^7 wt.% Pt/CMK-9 ^[e]^152 (at 250 °C)3 ^[f]^3 wt.% Pt/CMK-979 (at 250 °C)No deactivation was observed after 90 h TOS4 ^[f]^3 wt.% Pt-Fe/CMK-9114 (at 250 °C)5 ^[f]^7 wt.% Pt/3D-BMC-12 ^[g]^Fixed bed,45 bar,0.3 g catalyst,10 vol. % **ethylene glycol** in water,0.1 mL/min,WHSV = 2 h^−1^161 ^[c]^ (at 250 °C)No deactivation was observed after 25 h TOSWHSV is the weighted hour space velocity in g-feedstock/g-cat/h. ^[a]^ Data from Kim et al. [95]. ^[b]^ Pt surface area = 9.6 m^2^/g. Pt particle size = 2 nm. S_BET_ = 770 m^2^/g, V_pore_ = 0.87 cm^3^/g. Mesopore diameter = 4.1 nm. ^[c]^ mL/g/min was given by the authors. It was converted to mmol/g/h assuming 25 °C and 1 bar. ^[d]^ Data from Kim et al. [99]. ^[e]^ Pt surface area = 10.8 m^2^/g. Pt particle size = 1.6 nm. S_BET_ = 1713 m^2^/g, V_pore_ = 1.9 cm^3^/g. Mesopore diameter = 4.8 nm. ^[f]^ Data from the work of Park et al. [100]. ^[g]^ Pt particle size = 1.3 nm, S_BET_ = 1166 m^2^/g, V_pore_ = 1.57 cm^3^/g. Mesopore diameter = 4.7nm.


Using silica with interconnected pores as a hard template, Park et al. reported the use of three-dimensionally bimodal mesoporous carbon (3D-BMC-X, where X denotes the polymerization time of the carbon precursor) as a support in the APR of ethylene glycol (Table 13, Entry 5) [100]. The performance and stability of this material in the APR of ethylene glycol are comparable to those of other previously discussed ordered mesoporous carbons. Thermogravimetric and BET analyses of the spent catalysts revealed that the integrity of the support was maintained, and no Pt leaching was observed.

#### 2.5.3. Biomass-Derived Carbon Support

Gai et al. reported the synthesis of Ni nanoparticles supported on nitrogen-doped hydrochar (HC) with a unique thistle-like architecture for the APR of methanol [101]. The HC was synthesized through the hydrothermal carbonization of glucose, urea, and polyacrylate sodium (PAAS). One of the best catalysts, 2.2 wt.% Ni/HC-N_1_-S_1_ (where N_1_ refers to the mass of nitrogen doping via urea and S_1_ refers to the mass of PAAS used), was subjected to 10 recycling experiments. The freshly reduced catalyst exhibited a hydrogen production rate of 355 mmol/g-cat/h. After ten recycling cycles, a 16% decrease in the hydrogen production rate was observed (Table 14, Entry 1). Ni leaching into the solution was excluded as a source of activity loss. However, slight aggregation of Ni nanoparticles was noted, with sizes increasing from 5.7 nm to 6.9 nm after 10 cycles of methanol APR. Additionally, Raman spectroscopy suggested that the deposition of amorphous carbon could contribute to the observed loss in activity.

Chitosan, derived from the basic treatment of Chitin from the shells of crustaceans, was used by the group of Wang to synthesize N-doped carbon-supported Cu [98] or Ni [102] catalysts for the APR of methanol. Some of these works are highlighted in the following paragraphs.

Zheng et al. reported that Cu nanoparticles encapsulated in a carbon matrix derived from chitosan, synthesized by a sol–gel method, exhibited remarkable stability in the APR of methanol [98]. The catalyst, Cu@NC-200, carbonized at 200 °C under nitrogen, was found to be optimal. It maintained a hydrogen production rate of 34 mmol/g-cat/h during 200 h of continuous operation in the APR of methanol at 210 °C (Table 14, Entry 2). Transmission Electron Microscopy (TEM) analysis of the fresh and spent catalysts revealed no agglomeration of Cu nanoparticles, which remained predominantly between 5.7 and 5.8 nm.

Wu et al. reported the use of chitosan and glucose to encapsulate Cu nanoparticles [103]. The optimal catalyst, Cu@CS_19_-G_1_-300, was synthesized with a mass ratio of 19:1 for chitosan to *D*-glucose. The precursor was pyrolyzed at 300 °C, which was found to be the optimal temperature. At higher temperatures, the size of the Cu nanoparticles increased due to thermally induced aggregation. The stability of the catalyst in methanol APR at 210 °C was tested over five consecutive cycles, showing no significant decrease in H₂ production rate; the activity fluctuated between 131 and 136 mmol-H₂/g-cat/h (Table 14, Entry 3). Cu nanoparticle sizes remained close to 11.5 nm before and after the APR of methanol. XPS analysis revealed an increase in the amount of Cu⁰ after the reaction, at the expense of Cu⁺. They proposed that glucose enhances the reduction and dispersion of Cu species, leading to abundant Cu⁺/Cu⁰ interface sites that improve catalytic activity and stability.
molecules-29-04867-t014_Table 14Table 14Catalysts derived from biomass-related compounds in APR.EntryCatalystReaction ConditionH_2_ Prod. Rate (mmol/g-cat/h)Stability1 ^[a]^Ni/HC-N_1_-S_1_
^[b]^(2.2 wt.% Ni)Batch, 5 bar,0.2 g catalyst, 40 mL of 10 wt.% **methanol**1.5 h355 (at 250 °C)Estimated 16% loss (to 298) after ten cycles of 1.5 h each.5.5% loss in conversion 9.9% loss in H_2_ selectivity2 ^[c]^Cu@NC-200 ^[d]^(44.9 wt.% Cu)Fixed bed,40 bar,0.1 g catalyst,64 wt.% **methanol** in water,0.03 mL/min,WHSV = 15.8 h^−1^34 (at 210 °C)No significant loss after 200 h TOS.CO selectivity ≈ 0.03%3 ^[e]^Cu@CS_19_-G_1_-300 ^[f]^(35 wt/% Cu)Batch, 20 bar,0.03 g catalyst,10 mL of 37 wt.% **methanol** in water,1.25 h139 (at 210 °C)No significant loss are five cycles of 1.25 h each.H_2_ prod. rate fluctuated between 131 and 136.4 ^[g]^Ni@NC ^[h]^(40 wt.% Ni)Batch, 20 bar,0.025 g catalyst,10 mL of 25 mol.% **methanol** in water or 0.86 M KOH,1 h152 (at 220 °C in water)973 (at 220 °C, 0.86M KOH)4.2% loss (to 933) after nine cycles of 1 h each.5 ^[i]^Cu@Ca-Val-300 ^[j]^Fixed bed,20 bar,1 g catalyst,64 wt.% **methanol** in water,0.06 mL/min,WHSV = 3.22 h^−1^3 (at 180 °C)Stable for 110 h TOSWHSV is the weighted hour space velocity in g-feedstock/g-cat/h. ^[a]^ Data from Gai et al. [101]. ^[b]^ Ni dispersion = 38.7%, S_BET_ = 56 m^2^/g, V_pore_ = 0.086 cm^3^/g, Average pore diameter = 6.1 nm. ^[c]^ Data from Zheng et al. [98]. ^[d]^ S_BET_ = 7 m^2^/g, V_pore_ = 0.02 cm^3^/g, Average pore diameter = 15 nm. ^[e]^ Data from Wu et al. [103]. ^[f]^ d_Cu,XRD_ (fresh) = 11.3 ± 1.7 nm, d_Cu,XRD_ (spent) 11.5 ± 1.2 nm. ^[g]^ Data from Xiao et al. [102]. ^[h]^ Ni particle size (TEM) = 9 nm, S_BET_ = 32 m^2^/g, V_pore_ = 0.09 cm^3^/g, Average pore diameter = 11.6 nm. ^[i]^ Data from Li et al. [104]. ^[j]^ Cu dispersion = 42.4%, S_BET_ = 3.1 m^2^/g, V_pore_ = 0.008 cm^3^/g, Average pore diameter = 10.8 nm.


Xiao et al. reported the use of Ni nanoparticles encapsulated by a nitrogen-doped carbon framework (Ni@NC) derived from chitosan in the APR of methanol [102]. They reported that the hydrogen production rate of the freshly reduced catalyst is greatly enhanced by more than six times in the presence of KOH (Table 14, Entry 4). The catalyst lost about 4% of its initial hydrogen production rate after nine cycles of an hour each. Comparison of the X-ray diffraction (XRD) before and after methanol APR revealed no observable change in Ni particle sizes, which remained at approximately 9 nm. In the absence of KOH, Ni@CN produced a large amount of CO (32.3%) in the APR of methanol. This was reduced to 0.3% when the methanol APR was performed in an 0.86 M aqueous solution of KOH. They proposed that KOH reacted with CO to form potassium formate (KHCO_2_), thereby preventing the poisoning of the Ni by CO.

Li et al. reported carbon-encapsulated Cu catalyst in the APR of methanol [104]. The catalyst was prepared by the citric-acid-assisted sol–gel technique with *L*-valine. They found that the pyrolysis temperature had a significant influence on Cu particle size. The optimal catalyst Cu@Ca-Val-300 pyrolyzed at 300 °C has a Cu particle size of 14.3 ± 3.8 nm, estimated using TEM analysis. When the APR of methanol was performed in a batch setup, the hydrogen production rate was 97 mmol-H_2_/g-cat/h at a reaction temperature of 180 °C. In a fixed bed reactor, the activity was reduced to 3 mmol-H_2_/g-cat/h at the same reaction temperature (Table 14, Entry 5). The activity was found to increase for the first 90 h on stream before it stabilizes at 3 mmol-H_2_/g-cat/h for another 110 h on stream.

Chen et al. reported Cu nanoparticles encapsulated by N-doped carbon [105]. The N-doped carbon framework was derived from the pyrolysis of a precursor derived from the sol–gel synthesis of polyvinylpyrrolidone (PVP) and Cu nitrate. The optimal catalyst, Cu@NGC-600, had a high Cu loading of 68 wt.% and Cu particle size of 11.8 nm. The freshly reduced catalyst displayed a remarkable 166 mmol-H_2_/g-cat/h at 190 °C. As no stability data was provided, this catalyst is not included in Table 14.

#### 2.5.4. Carbon-Encapsulated Metal Oxide Support

Carbon encapsulation could potentially confer stability on the metal nanoparticles by minimizing migration and coalescence which will cause the sintering of metal nanoparticles, especially at high reaction temperatures [106]. In this section, we will explore the use of carbon-encapsulated catalysts in the aqueous-phase reforming of alcohols, and whether encapsulation indeed improves the catalyst stability in this context.

Zheng et al. reported the use of an N-doped carbon-encapsulated Cu/ZnO catalyst, labeled as Cu/ZnO@NC, for the aqueous-phase reforming (APR) of methanol in the presence of KOH [107]. The carbon encapsulation improved the hydrophilicity of the catalyst, measured by the contact angle of water on the catalyst’s surface, which correlated with the rate of hydrogen production via methanol APR. The encapsulation also significantly enhanced the hydrothermal stability of the catalyst. In comparison to the control catalyst, Cu/ZnO, which lost 81% of its initial activity, Cu/ZnO@NC only lost 17% of its initial activity (Table 15, Entry 1 vs. 2). Analysis of the fresh and spent catalysts revealed significant aggregation of Cu nanoparticles in the absence of carbon encapsulation. The Cu particle sizes in Cu/ZnO increased from 4.5 nm to 47.2 nm, while those in Cu/ZnO@NC increased from 3.8 nm to 7.8 nm. Some leaching of Cu possibly occurred as the mass content in both catalysts decreased by 2–5% after the reactions. XRD analysis showed that the ZnO in Cu/ZnO was hydrothermally transformed to β-Zn(OH)₂, whereas in Cu/ZnO@NC, the support remained intact.

The same group reported carbon-encapsulated Cu/Al_2_O_3_-ZnO in the APR of methanol [108]. The carbon source was Sesbania powder (SP). The resulting catalyst had a petal-like hollow morphology. The carbon-encapsulated catalyst, Cu-SP/Al_2_O_3_-ZnO, demonstrated an enhanced hydrogen production rate compared to the commercial Cu/ZnO/Al_2_O_3_ catalyst (Table 15, Entry 3 vs. 4). While it exhibited less activity loss after five cycles of methanol APR relative to Cu/ZnO/Al_2_O_3_, it did lose 44% of its initial activity (Entry 4). Cu-SP/Al_2_O_3_-ZnO experienced a 5.1 wt.% loss in Cu after the reaction, which is much lower than the 27.6 wt.% loss of Cu in Cu/ZnO/Al_2_O_3_. Aggregation of Cu nanoparticles was observed in both catalysts, but it was less severe in Cu-SP/Al_2_O_3_-ZnO (fresh: 11.3 nm, spent: 13.6 nm) than in Cu/ZnO/Al_2_O_3_ (fresh: 15 nm, spent: 31 nm).

#### 2.5.5. Carbon Nanotubes/Fibers

Carbon nanotubes are a highly structured form of carbon with exceptional mechanical strength and electrical conductivity compared to activated carbon [109]. However, they generally have a lower specific surface area compared to activated carbon. Nevertheless, their tunability in terms of elemental and surface composition to modify catalytic properties makes them an attractive choice [110]. In this section, we will highlight the use of carbon nanotubes or fibers in the aqueous-phase reforming of alcohols.

Van Haasterecht et al. reported on the aqueous-phase reforming (APR) of ethylene glycol (EG) using Pt or Ni supported on carbon nanofibers (CNF) [111]. Using a catalyst from their previous work (5 wt.% Pt/CNF) [112], they demonstrated that Pt/CNF exhibited both high performance and stability in the APR of EG. The hydrogen production rate of Pt/CNF reached 39.6 mmol-H_2_/g-cat/h at 230 °C, with no significant loss in activity observed after over 50 h on stream (Table 16, Entry 1).

They found that the stability of Ni/CNF increased significantly when the APR of EG was performed with KOH. In the absence of KOH, the hydrogen production rate of Ni/CNF peaked at 12 mmol-H_2_/g-cat/h after 2 h on stream, with about 92% of this activity lost by the 50 h mark (Table 16, Entry 2). However, when KOH was added to the APR of EG, the activity increased to 25.5 mmol-H_2_/g-cat/h and was maintained for 50 h on stream.

Comparative analysis between the freshly reduced and spent Ni/CNF revealed that the aggregation of Ni particles and the leaching of Ni into the solution were significantly suppressed when the APR of EG was conducted in the presence of KOH. The authors attributed the increase in stability primarily to the suppression of Ni particle aggregation. While Ni particle sizes increased from 8 nm to 58 nm without KOH, the increase was less drastic with KOH (12 nm in the spent catalyst). The leaching of Ni into the solution was minimal in both cases, with less than 0.3 wt.% loss of Ni via leaching.

Pioneering work by Haller’s group established the high performance of Pt and Pt-Co catalysts supported on both single- and multi-walled carbon nanotubes in the APR of ethylene glycol [113,114,115,116]. The APR stability of carbon-nanotube-supported catalysts was subsequently explored by other groups and will be discussed below.

Building on the precedent of Cu-Ni alloy as a superior catalyst compared to Ni alone in the steam reforming of methanol, Rahman reported on the use of bimetallic Cu and Ni supported on multi-walled carbon nanotubes (MWNT) as catalysts for the aqueous-phase reforming (APR) of glycerol [117]. They found that the MWNT-supported Ni catalyst (12Ni/MWNT) lost all its activity after 75 h on stream. In contrast, the catalyst with both Cu and Ni (1Cu-12Ni/MWNT) remained stable for 110 h (Table 16, Entry 3 vs. 4). They observed that the crystalline particle sizes of Ni in 12Ni/MWNT increased from 15.8 nm to 21.7 nm after 110 h on stream, while the particle size of Ni in 1Cu-12Ni/MWNT showed a smaller increase of from 9.8 nm to 11.3 nm. In addition to improving the stability of the catalyst under APR conditions, 1Cu-12Ni/MWNT exhibited a higher H_2_ yield and lower CH_4_ and CO yields compared to 12Ni/MWNT.
molecules-29-04867-t016_Table 16Table 16Carbon-nanotube/fiber-supported catalysts in the APR of alcohols.EntryCatalystReaction ConditionH_2_ Prod. Rate (mmol/g-cat/h)Stability1 ^[a]^Pt/CNF ^[b]^(5 wt.% Pt)Fixed bed,29 bar,0.1 g catalyst,10 wt.% **ethylene glycol** (EG) in water,0.05 mL/minWHSV = 3g-EG/g-cat/h39.6 ± 0.5 (at 230 °C)No observable loss in activity after 50 h TOS2 ^[a]^Ni/CNF ^[c]^(12.5 wt.% Ni)*with 0.5M KOH*: 25.5 ± 0.8 (at 230 °C)*With 0.5M KOH:* No observable loss in activity after 50 h TOS*No KOH*: 12 (at 230 °C at 2h TOS)*No KOH:* 92% loss (to 1) in activity after 50 h TOS3 ^[d]^12Ni/MWNT ^[e]^(12 wt.% Ni)Fixed bed,40 bar,0.15 g catalyst,1 wt.% **glycerol** in water,0.05 mL/minWHSV = 20 h^−1^6.2 ± 0.4 (at 240 °C) ^[f]^100% activity loss by 75 h TOS4 ^[d]^1Cu-12Ni/MWNT ^[g]^(1 wt.% Cu,12 wt.% Ni)9.5 ± 0.4 (at 240 °C)Stable for 110 h TOS5 ^[h]^5Pt-1.5Ni/MWNT ^[i]^ (4.7 wt.% Pt, 1.4 wt.% Ni)Batch,30 bar,0.1 g catalyst,15 mL of 10 wt.% **glycerol** in water,4 h reaction*no CaO*: 9.1 (at 230 °C)N/A*With CaO:* 18 (at 230 °C)*With CaO:* 26% loss in activity ^[j]^ after five cycles of 4 h each.WHSV is the weighted hour space velocity in g-feedstock/g-cat/h unless otherwise stated. ^[a]^ Data from van Haasterecht et al. [111]. ^[b]^ Characterization data from van Haasterecht et al. [112]. Average Pt particle sizes = 3 nm, S_BET_ = 169 m^2^/g, V_pore_ = 0.31 cm^3^/g. ^[c]^ Average Ni particle size = 8 nm. No textual properties given in van Haasterecht et al. [111]. ^[d]^ Data from Rahman. [117]. ^[e]^ Ni dispersion = 6.4%, S_BET_ = 260 m^2^/g, V_pore_ = 1.38 cm^3^/g, Average pore diameter = 31.1 nm. ^[f]^ 40 h averaged values. ^[g]^ Metal dispersion = 10.3%, S_BET_ = 273 m^2^/g, V_pore_ = 1.45 cm^3^/g, Average pore diameter = 31.0 nm. ^[h]^ Data from He et al. [118]. ^[i]^ Ni dispersion = %, S_BET_ = 141 m^2^/g, V_pore_ = 1.29 cm^3^/g, Average pore diameter = 36.7 nm. ^[j]^ Based on H_2_ yield (mmol-H_2_/g-glycerol).


He et al. reported on the use of bimetallic Pt and Ni supported on multi-walled carbon nanotubes as catalysts for the APR of glycerol in the presence of CaO [118]. The hydrogen production rate of the freshly reduced catalyst nearly doubled, from 9.1 to 18 mmol-H_2_/g-cat/h, when CaO was added to the reactor (Table 16, Entry 5). The addition of CaO also significantly reduced CH_4_ formation, from 40.4% to 0.21%, and concurrently increased H_2_ selectivity, from 33.2% to 59.4%. This improvement is attributed to CaO facilitating the water–gas shift reaction and inhibiting methanation through in situ CO_2_ sorption via carbonation, thereby enhancing H_2_ selectivity and reducing CH_4_ formation. However, the authors noted that CaO caused the deactivation of the catalyst, which can be regenerated by pyrolysis at 750 °C. In a five-cycle stability test, the optimal catalyst, 5Pt-1.5Ni/MWNT, lost 26% of its initial activity.

Tang et al. reported the APR of glycerol with porous carbon nanofiber (PCNF)-supported Ni catalysts [119]. The PCNFs were fabricated using the electrospinning technique. The encapsulation of Ni nanoparticles within porous carbon nanofibers (Ni@PCNF) further improved catalyst stability, leading to high-purity hydrogen production (93%) with minimal byproducts such as CH_4_ and CO. However, as we are unable to extract stability data from their work, it is not included in Table 16.

### 2.6. Molybdenum Carbide or Sulfide

Atomically dispersed Ni and Pt on α-Mo_2_C are prominently featured as catalysts with high activities for the aqueous-phase reforming (APR) of methanol [40,80]. These catalysts achieved some of the highest specific hydrogen production rates for methanol APR in the literature. A 2 wt.% Pt/α-Mo2C catalyst produced hydrogen at a rate of 467 mmol/g-cat/h at 190 °C. However, after 11 cycles of methanol APR, it lost 33% of its initial activity (Table 17, Entry 1). The authors attributed this loss in activity to catalyst loss due to batch reactor stirring, which caused some of the catalyst to be dislodged, and the accumulation of CO_2_ in the liquid. Similarly, at the same loading, Ni/α-Mo_2_C exhibited a hydrogen production rate of 626 mmol/g-cat/h at 240 °C (Table 17, Entry 2) and experienced a similar degree of deactivation, losing 30% of its initial activity after 10 cycles of methanol APR.

Liu et al. reported on the performance of a MoS_2_-nanosheet-supported Pt catalyst in the APR of methanol under basic conditions [37]. The number of layers in the MoS_2_ nanosheet could be controlled by the temperature and duration of the hydrothermal synthesis, with a nanosheet of six layers identified as the optimal support. At a 0.2 wt.% Pt loading, the Pt/MoS_2_ catalyst achieved a hydrogen production rate of 11.5 mmol-H_2_/g-cat/h (Table 17, Entry 3). In contrast, bulk MoS_2_ with the same Pt loading showed only 58% of the performance of the nanosheet-supported catalyst. The catalyst lost 24% of its initial activity after four cycles of methanol APR. From TEM analysis, they attributed the deactivation to the aggregation of Pt nanoparticles.

## 3. Discussion—Sonolysis of Aqueous Alcohol for Hydrogen Production

In catalytic thermal aqueous-phase reforming of alcohols, thermal energy is directly supplied by heating elements, while the catalyst lowers activation barriers, enabling faster reactions. In contrast, ultrasound can induce alcohol reforming through extreme thermal conditions generated by acoustic cavitation (see Section 3.1 for further details). In the absence of a catalyst, the efficiency of hydrogen production from aqueous alcohols is influenced by several factors, which are discussed in Section 3.2. Mechanistic insights into ultrasound-induced sonolysis of alcohols are provided in Section 3.3. The role of piezocatalysts, which convert mechanical energy into electrical energy to drive chemical reactions, and sonocatalysts, which enhance the efficiency of acoustic cavitation, is covered in Section 3.4.

The ability of ultrasound to induce the decomposition of water and aqueous alcohols has been extensively studied and documented in the literature. The sonolysis of alcohols has been thoroughly examined by Dehane et al. [120]. Therefore, we will not delve into all the known variables that influence the outcomes of their sonolysis. Instead, we will focus on fundamental parameters such as ultrasound frequency, purging gas (including its composition, which can have a strong impact on sonochemistry [121,122]), and power input/acoustic intensity where applicable. Additionally, the use of sonocatalysts and piezocatalysts to enhance the sonolysis of alcohols in water will be discussed, as these have the potential to significantly increase hydrogen production via the sonolysis of aqueous alcohols beyond the current limit.

Other authoritative reviews focusing on different facets of sonochemistry are available in the literature. For an in-depth discussion on sono-reactor design to optimize the sonolysis of aqueous methanol, including theoretical models and computational fluid dynamics (CFD) techniques, please refer to the work of Dehane et al. [120]. A thorough examination of theoretical aspects, reactor engineering, and scaling-up considerations can be found in the study by Meroni et al. [123]. Merabet and Kerboua reviewed the state-of-the-art in sonocatalysis, sonoelectrocatalysis, sonophotolysis, and sonophotocatalysis for hydrogen production, emphasizing the importance of controlling parameters and reactor design for efficiency [124]. On a more focused note, Islam et al. discussed the potential of using sonochemical and sonoelectrochemical methods to produce clean hydrogen efficiently, highlighting how these processes enhance traditional water electrolysis [125]. Kiss et al. highlighted ultrasound-assisted emerging technologies in chemical processes, detailing advancements in extraction, crystallization, and reactive distillation, along with a proposed roadmap for the industrial implementation of these technologies [126]. Wood et al. detailed the key factors influencing sonochemical activity in aqueous solutions, including primary parameters such as pressure amplitude, frequency, and reactor design, and secondary parameters like the use of gas and liquid additives, highlighting their effects on bubble dynamics and sonochemical efficiency [127].

### 3.1. Basic Theoretical Background

The application of ultrasound to a liquid medium results in the formation of tiny gas bubbles, which expand and eventually collapse. This process is known as acoustic cavitation [128]. The implosion of bubbles can result in localized hotspots with temperatures ranging from a few thousand to tens of thousands Kelvin and pressures of a few hundred to a few thousand bars [129,130]. Reactive radicals can be produced inside these hotspots (primary sonochemistry) or outside these hotspots (secondary sonochemistry) [131]. Heating and cooling due to acoustic cavitation occur at a very high rate (in the order of picoseconds), thus enabling unique chemistry to occur via ultrasonic activation that would otherwise be not possible under an ambient condition.

The occurrence and size of these bubbles, which depend on the frequency of the applied ultrasound, will determine whether physical or chemical effects dominate. For low-frequency ultrasound (20–100 kHz), physical effects such as liquid circulation and turbulence dominate, as cavitation events are much rarer compared to higher-frequency ultrasound. Despite producing larger acoustic cavities, lower-frequency ultrasound is not optimal for radical yield. Radical yield is a balance between the size of the bubbles and their occurrence, and bubble occurrence increases with increasing frequency. Therefore, the optimal range for radical yield is between 200 and 600 kHz [123].

For detailed theoretical and experimental aspects of acoustic cavitation, readers are referred to the reviews that are cited in this section.

### 3.2. Sonolysis of Aqueous Alcohols

It is known that the presence of low concentrations of methanol in water can highly enhance the hydrogen production rate via sonolysis. This has been consistently reported by many reports. In this section, we will discuss various works on hydrogen production via the sonolysis of aqueous alcohols.

The attenuation of ultrasound results in a lower acoustic intensity as the distance from the transducer is increased (Figure 7) [123]. In proof-of-concept studies, sono-reactors are usually limited to one transducer with no optimization of configuration. Therefore, to mitigate the impact of the liquid volume on the hydrogen production rate, we employed a volume-normalized metric. However, we note that optimization of transducer configuration is an essential component in the design of sono-reactors, therefore, we have included the liquid volume used so that the reader can calculate the reported hydrogen production rate easily by multiplying with the values given in µmol/mL-liquid/h.

Buttner et al. studied the sonolysis of aqueous methanol at 1 MHz [133]. Under argon, they found that the optimal amount of methanol was around 8 wt.% in water. The hydrogen production rate at this concentration was approximately 9 µmol/mL-liquid/h. Compared to sonolysis of 8 wt.% methanol in water, at 35 wt.%, the selectivity for formaldehyde and CH_4_ increased significantly from 15% and 9% to 30% and 15%, respectively, with a concomitant decrease in H_2_ selectivity. When the sonolysis was performed in oxygen, CO_2_, which is formed from combustion, was the main product.

Rassokhin et al. explored the hydrogen evolution rate of the sonolysis of aqueous methanol at various temperatures [134]. They employed intermediate frequency ultrasound at 724 kHz. They found that hydrogen production was greatly accelerated by adding a small amount of methanol to water (Figure 8). The hydrogen production rates for water and methanol individually were low, generally less than 1 µmol/mL-liquid/h. However, if 3.2 wt.% of methanol was added, a hydrogen production rate of 12 µmol/mL-liquid/h could be achieved over a range of temperatures (38–50 °C). The change in reaction pathways between the sonolysis of water and that of a water–methanol mixture is further discussed in Section 3.3.

Mizukoshi et al. reported the sonolysis of various C1-C6 alcohols at 5 °C [135]. The results are depicted in Figure 9. Intermediate-frequency ultrasound of 200kHz and an acoustic intensity of 6 W/cm^2^ were used. The sonolysis of ethanol and 1-propanol produced hydrogen at a rate of 2.4 and 3.0 µmol/mL-liquid/h, respectively, which is a drastic improvement over methanol (1.3 µmol/mL-liquid/h). However, the amount of CH_4_ produced also increased for ethanol and 1-propanol over methanol. Interestingly, the branched isomer of 1-propanol—2-propanol—appeared to have a very high resistance to sonolysis, which was only exceeded by that of pentanol and hexanol. They concluded that for alcohols, the evaporation rate rather than vapor pressure is a better descriptor for sonolysis activity. This is in contrast to the sonolysis of hydrocarbons, where vapor pressure demonstrates a good negative correlation with sonolysis rate [136,137].

### 3.3. Insights into the Mechanism of Sonolysis

Spin-trapping experiments and electron spin resonance spectroscopy have been used to provide insights into the radical species formed during the sonolysis of aqueous methanol. Krishna et al. reported the observation of CH_3_ and CH_2_OH radicals from the sonolysis of aqueous methanol at 50 Hz [138]. The concentration of these radicals peaks at 5 M of methanol in water (about 16 wt.% methanol in water) and decreases drastically with increasing amounts of methanol. They rationalized this observation by the decrease in temperature that the collapse of the bubble can reach due to the higher heat capacity of methanol vapor compared to water vapor, and the increase in thermal conductivity with increasing methanol content. A decrease in bubble temperatures with increasing alcohol concentration has also been reported by Rae et al. [139].

Yasui studied the non-equilibrium reactions and physical properties of the acoustic cavitation of water in the presence of argon via numerical simulation [121]. To the left of Figure 10, the relationship between the bubble radius and temperature during the final stage of bubble collapse is shown. As the bubble collapses violently in a matter of picoseconds, the temperature inside rises sharply, reaching around 5100 K. This increase in temperature is primarily due to the work done by the liquid surrounding the bubble. After reaching this peak, the temperature decreases rapidly as the bubble expands. At the same time, there is a dramatic increase in pressure, reaching approximately 6 GPa, which leads to a sudden halt in the bubble collapse (Right of Figure 10). This halt occurs when the density inside the bubble nears that of a condensed phase, which occurs at around 650 kg/m^3^.

Kerboua and Hamdaoui conducted numerical simulations on a single acoustic bubble to investigate the impact of varying dioxygen and argon compositions on the production rates of various chemical species during the sonolysis of water and methanol–water mixtures [140]. In pure water, a molar fraction of 90% argon resulted in an optimal sonolysis rate due to peak bubble temperatures and increased radical lifetimes afforded by the high concentration of inert argon. Both oxygen atoms and hydroxyl radicals are prevalent at both high and low argon compositions relative to dioxygen. However, at high argon compositions, hydrogen radicals become more prominent, while at low argon but high oxygen compositions, the HOO· radical is more prevalent. When methanol is added to water, the chemical dynamics within the bubble change significantly, giving rise to new species such as CH_3_O, CH_2_O, HCO, CO_2_, and CO, which are not present in pure water sonolysis. For a 1% (*v*/*v*) methanol–water mixture, with a bubble containing a high oxygen level (90% molar oxygen), CO and H_2_O_2_ emerge as the major products, followed by CH_2_O and H_2_. At this oxygen–argon composition, the production of H_2_ is significant but not necessarily at its maximum, which tends to occur with different methanol concentrations. Conversely, at 90% molar argon, species such as CO, ·O, ·OH, and CO_2_ are drastically reduced, indicating that methanol not only enhances the selectivity of sonochemical reactions but also shifts the chemical pathways to produce different species compared to pure water.

### 3.4. Sonocatalyst or Piezocatalyst

Solid cavitation agents can enhance acoustic cavitation efficiency and lower the energy consumption of a reaction driven by ultrasound [141]. The use of nanostructured AuPd/TiO_2_ as sonocatalysts for the oxidation of benzyl alcohols has been reported [142]. However, the application of these cavitation agents in the sonolysis of aqueous alcohols is rare.

Wang et al. reported that Au/TiO_2_ significantly improved the hydrogen production rate via the sonolysis of various alcohols, including methanol, ethanol, and glycerol, in water [143]. Au/TiO_2_ increased the hydrogen production rate of water and aqueous methanol sonolysis by about 53-fold (Table 18, Entries 1 and 2) and about 83-fold (Entries 3 and 4), respectively, compared to sonolysis performed without Au/TiO_2_. For the sonolysis of 1.9% (*v*/*v*) ethanol in water, Au/TiO_2_ increased the hydrogen production rate by 248 times, achieving a value of 1.3 µmol/mL-liquid/h. The effect on 0.67% (*v*/*v*) glycerol in water was less pronounced, with a 31-fold increase, resulting in a hydrogen production rate of 0.2 µmol/mL-liquid/h. For comparison, the specific hydrogen production rate from the sonolysis of aqueous methanol is 3.8 mmol/g-cat/h, which is noteworthy compared to the thermal aqueous-phase reforming of methanol since no room temperature APR via thermal catalysis has been reported thus far. However, the scalability of this technology remains to be evaluated.

Zhang et al. reported that the sonolysis of a suspension of BaTiO_3_ in 10 wt.% methanol in water could produce hydrogen at a rate of 6 mmol/g-cat/h over a duration of 4 days [132]. The hydrogen production rate, normalized by liquid volume, was 0.6 µmol/mL-liquid/h (Table 18, Entry 5) with 10 mg of BaTiO_3_ as a piezocatalyst. They found it crucial to disperse the BaTiO_3_ via mechanical stirring or agitation before the sonolysis; otherwise, negligible hydrogen would be produced. Additionally, they observed that the BaTiO_3_-catalyzed sonolysis of aqueous methanol was sensitive to various factors, including the vertical distance of the reactor to the transducer, the catalyst dosage, and the volume of liquid used.
molecules-29-04867-t018_Table 18Table 18Compilation of selected results on the sonolysis of aqueous methanol.EntryFrequency (kHz)Power (W)CatalystMethanol in Water (Wt.%)Optimal Temp. (°C)Liquid Vol. (mL)H_2_ Prod. Rate (µmol/mL-liquid/h) ^[a]^14050
021–251500.002724050Au/TiO_2_
^[b]^021–251500.14434050
421–251500.02344050Au/TiO_2_
^[b]^421–251501.954060BaTiO_3_ ^[c]^8.1351000.662006 W/cm^2^
1005101.3772445
038–50400.7872445
3.229–474012972445
100−7400.41010002 W/cm^2^
0N/A401.41110002 W/cm^2^
7N/A409.2Data in entries 1–4 are from Wang et al. [143]. Entry 5 data are from Zhang et al. [132]. Entry 6 data are from Mizukoshi et al. [135]. Data in entries 7–9 are from Rassokhin et al. [134]. Data in entries 10 and 11 are from Büttner et al. [133]. ^[a]^ For works that reported H_2_ production rate in the units of µM/min, we assumed that the volume refers to the volume of liquids used in the reaction. This is converted to µmol/min based on the volume of liquid reactants used in the reaction. ^[b]^ 0.075g of 0.75 wt.% Au/TiO_2_. Au particle size (TEM) = 3.5 ± 0.7 nm. ^[c]^ 0.01 g of BaTiO_3_ was used.


While comparing results from different reports can be misleading as frequency may not be the only significant factors that affect the rate of hydrogen production from an alcohol–water mixture, it could be nonetheless instructive to compile results from the sonolysis of water, methanol, and their mixture (Table 18). Intermediate-frequency ultrasound, at a similar power input, seems to be superior to lower-frequency ultrasound in the production of H_2_ from sonolysis. For instance, at 724 kHz, the sonolysis of 3.2 wt.% methanol in water is reported to produce 12 µmol/mL-liquid/h or 480 µmol/h in 40 mL of 3.2 wt.% methanol in water (Entry 8). The use of 1000 kHz ultrasound seems to be comparable for the sonolysis of aqueous methanol (Entry 11) This is remarkable considering that low-frequency ultrasound produced 0.023 µmol/mL-liquid/h or 3.5 µmol/h in 150 mL of 4 wt.% methanol in water (Entry 3). A similar trend can be seen for the sonolysis of water (Entries 1, 7, and 10).

## 4. Conclusions

In this review, we have surveyed the literature on the aqueous-phase reforming (APR) of model alcohols such as methanol, ethanol, ethylene glycol, and glycerol. The APR of these alcohols is mainly catalyzed by Pt, Ni, Co, or their binary mixtures, and supported on metal oxides, carbon, Mo_2_C, or MoS_2_. State-of-the-art catalysts are currently dominated by the use of expensive Pt as the active metal. Pt-based catalysts have been demonstrated in various works to be stable for extended duration under APR (up to 600 h in the case of Pt/NiAl_2_O_4_). Much research has been carried out on the use of more cost-effective metals such as Ni, Cu, and Co, which has resulted in some promising catalysts with room for improvement.

In terms of stability, it appears that there is still a significant gap between laboratory-based research and industry-based application. One of the recommended metrics for industry is catalyst consumption (kg-cat per tons-product) [18]. The ideal value for catalyst consumption is less than 0.1, which implies in this case, that for 1 g of catalyst, 10 kg (or 5 kmol) of H_2_ has to be produced. For APR catalysts tested under continuous operation in a fixed bed reactor, the specific hydrogen production rate ranges from 26–160 mmol/g-cat/h. If we take the median of 90 mmol-H_2_/g-cat/h, which translates to 0.18 g-H_2_/g-cat/h or 4.3 g-H_2_/g-cat/day, the catalyst will have to run for more than 2300 days to meet this criterion. Therefore, we envisaged further research is needed to achieve this.

The ultrasound-induced sonolysis of water and aqueous alcohols was extensively studied and reported in the 1980s; however, commercially viable hydrogen generation via this technology remains out of reach. Promising works on the use of piezocatalysis or sonocatalysis to enhance the efficiency of sonolysis have been demonstrated; however, further research would be required to push the boundary of sonolysis of aqueous methanol before it can reach a higher technological readiness level.

In conclusion, while hydrogen is a clean and efficient fuel, its widespread use is limited by logistical challenges. Liquid organic hydrogen carriers, such as alcohols, present a viable solution through aqueous-phase reforming (APR), which eliminates the need for energy-intensive feedstock vaporization. However, the success of APR on an industrial scale hinges on addressing the stability of catalysts under hydrothermal conditions, which is critical for sustained hydrogen production. Furthermore, ultrasound-assisted APR offers a promising alternative by enabling hydrogen production without external heating, although this technology is less advanced compared to thermal APR. The continued research and development of catalysts specifically designed for ultrasound-assisted processes could open new avenues for efficient hydrogen production from alcohols.

## Figures and Tables

**Figure 1 molecules-29-04867-f001:**
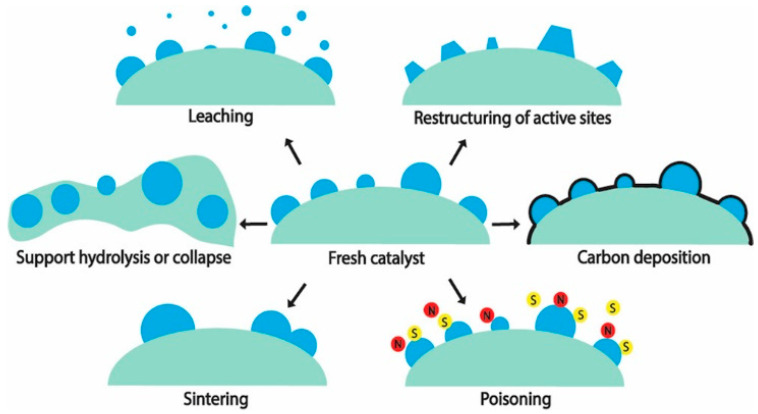
Deactivation mechanisms for heterogeneous catalysts under hydrothermal conditions. Reprinted with permission from ref. [21]. Copyright 2021 American Chemical Society.

**Figure 2 molecules-29-04867-f002:**
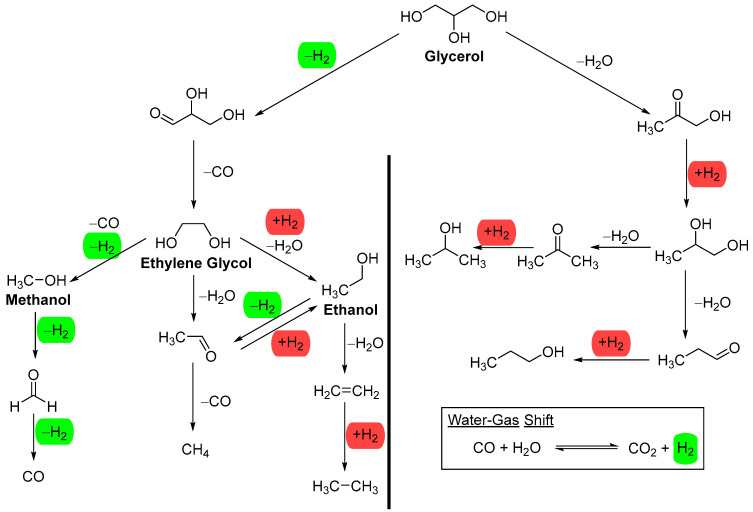
Reaction pathways for the aqueous-phase reforming of glycerol, ethylene glycol, ethanol, and methanol. On the left side of the image are reactions that produce and consume H_2_, while on the right side are reactions that only consume H_2_, except for the water–gas shift reaction which produces H_2_ from CO and water. Green: H_2_ produced from reaction. Red: H_2_ consumed in reaction.

**Figure 3 molecules-29-04867-f003:**
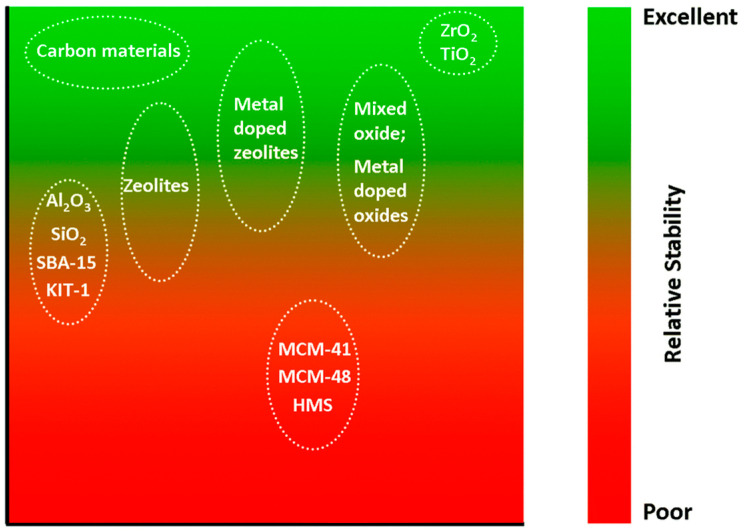
Hydrothermal stability of supports for heterogeneous catalysts in hot water at less than 374 °C. Reprinted from ref. [32] with permission. Copyright 2014 Royal Society of Chemistry.

**Figure 4 molecules-29-04867-f004:**
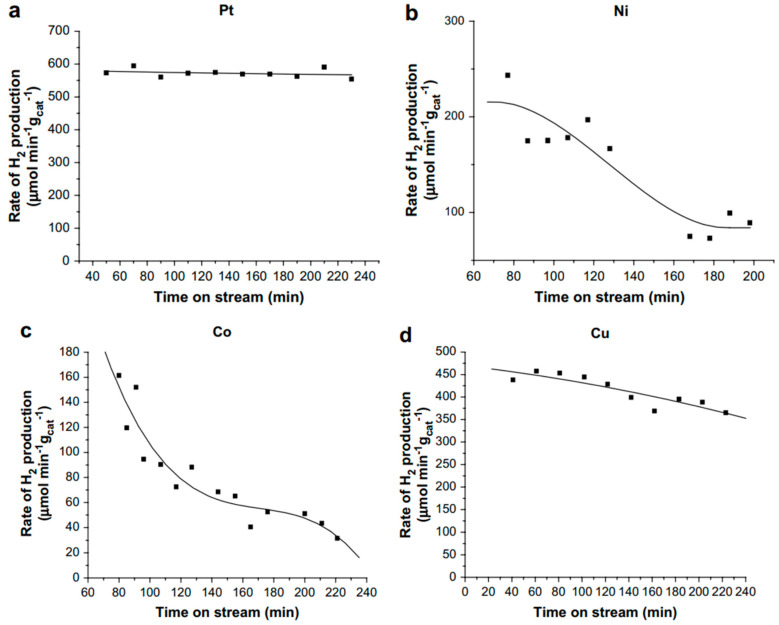
Stability of various active metals supported on alumina in glycerol APR at 230 °C and 32 bar. (**a**) 4.4 wt.% Pt/Al_2_O_3_. (**b**) 17.4 wt.% Ni/Al_2_O_3_ (**c**) 15.38 wt.% Co/Al_2_O_3_ (**d**) 6.1 wt.% Cu/Al_2_O_3_. Reprinted from International Journal of Hydrogen Energy, 33, Wen, G.; Xu, Y.; Ma, H.; Xu, Z.; Tian, Z., Production of hydrogen by aqueous-phase reforming of glycerol, 6657, Copyright (2008), with permission from Elsevier.

**Figure 5 molecules-29-04867-f005:**
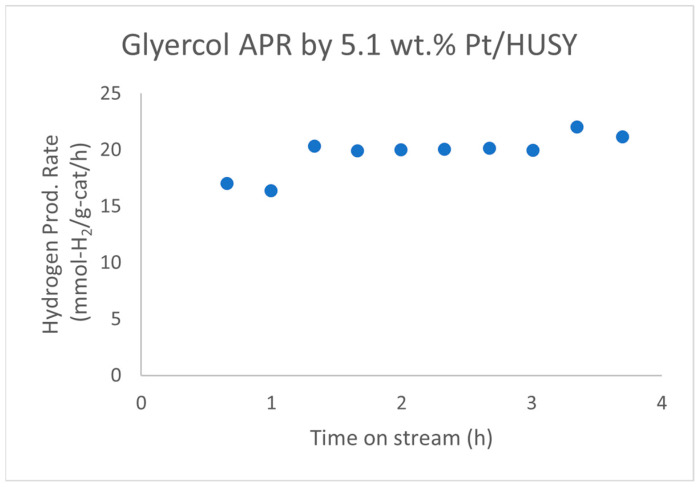
Hydrogen production rate of Pt/HUSY in the APR of 10 wt.% glycerol in water at 230 °C, 32 bar, LHSV of 8 h^−1^. Data extracted from the work of Wen et al. [42] with WebPlotDigitizer [58].

**Figure 6 molecules-29-04867-f006:**
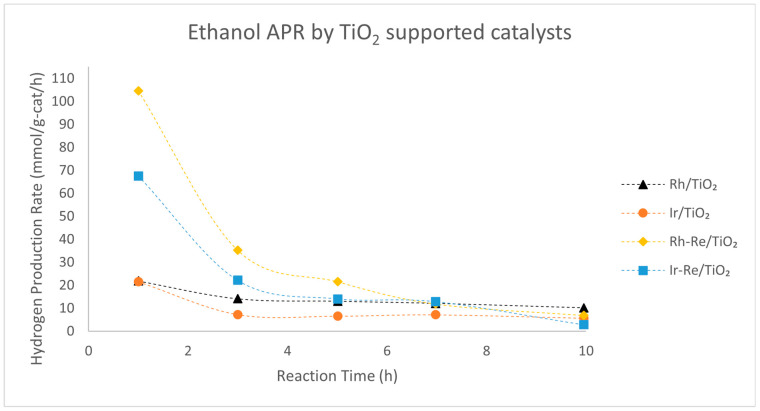
Data extracted from Nozawa et al. [82] with WebPlotDigitizer [58]. Reaction conditions: 0.5 g of catalysts, 80 mL of 10 vol.% aqueous EtOH, reaction temperature = 200 °C and initial pressure = 20 bars. 5 wt.% Rh/TiO_2_, 5 wt.% Ir/TiO_2_, 5 wt.% Rh–5 wt.% Re/TiO_2_ and 5 wt.% Ir–5 wt.% Re/TiO_2_. EtOH conversion after 10 h for Rh/TiO_2_ = 14.2%, Ir/TiO_2_ = 5.9%, Rh-Re/TiO_2_ = 36.1% and Ir-Re/TiO_2_ = 14.7%.

**Figure 7 molecules-29-04867-f007:**
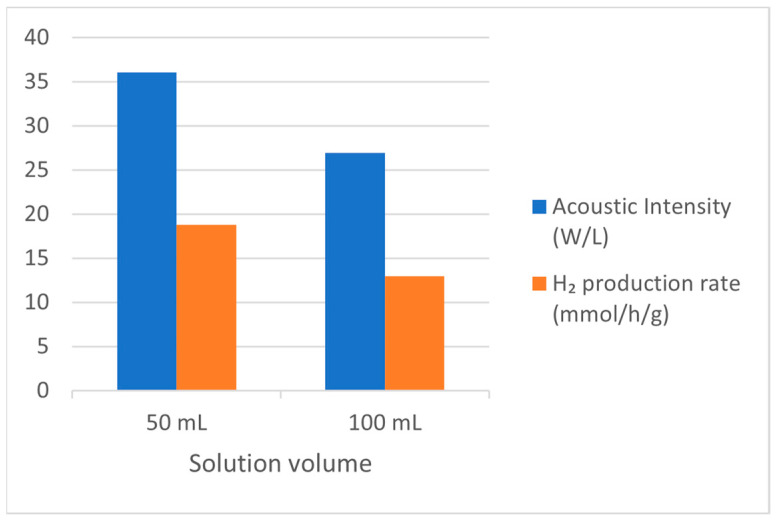
The dependency of acoustic intensity and hydrogen production rate on liquid volume. Data from Zhang et al. [132].

**Figure 8 molecules-29-04867-f008:**
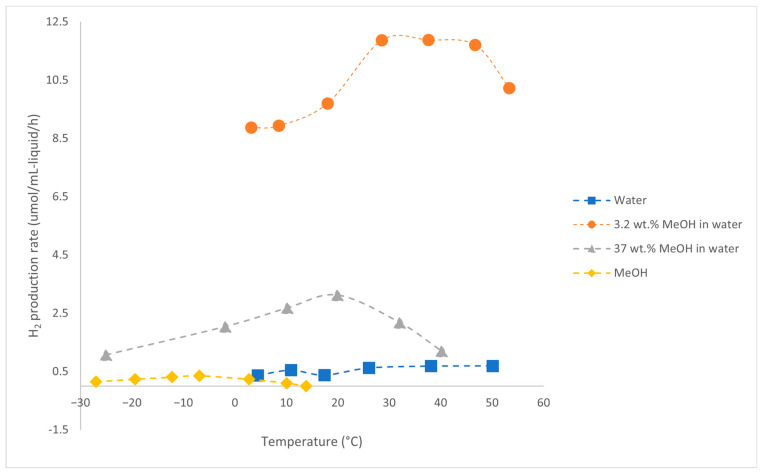
Temperature-dependent hydrogen production rate from the sonolysis of water, methanol, and aqueous methanol at 724 kHz as reported by Rassokhin et al. [134]. The reaction volume of the liquid was 40 mL.

**Figure 9 molecules-29-04867-f009:**
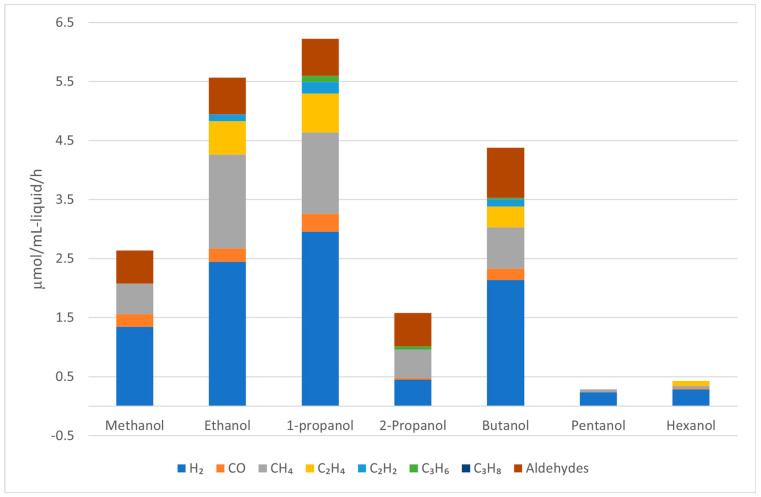
Sonolysis of various alcohols at 5 °C. The liquid volume used was 10 mL. Data are extracted from Mizukoshi et al. [135].

**Figure 10 molecules-29-04867-f010:**
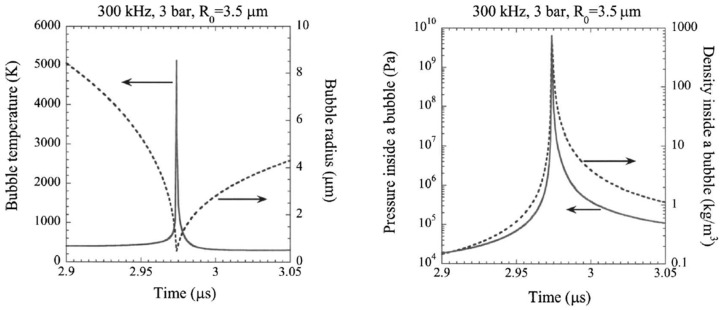
**Left**: Bubble temperature and radius around the vicinity of the bubble collapse. **Right**: Pressure and density inside of the bubble during the collapse of the bubble. Reprinted from *J. Chem. Phys.* 2007, 127 (15), 154502., with the permission of AIP Publishing.

**Table 1 molecules-29-04867-t001:** Saturated vapor pressure of model alcohols considered in this review.

Alcohols	wt.% (mol.%) in Water	Temperature (°C)		
		200	225	250
Methanol	10 (5.9)	18.5	29.9	46.2
	50 (36)	27.4	43.5	66.3
Ethanol	10 (4.2)	18.3	29.5	45.4
	50 (28)	25.5	40.6	61.5
Ethylene Glycol	1 (0.3)	15.5	25.4	39.7
	10 (3.1)	15.2	24.9	38.8
Glycerol	1 (0.2)	15.5	25.5	39.7
	10 (2.1)	15.2	24.9	38.8

Calculated with DWSIM 8.8.1 using the Peng–Robinson–Stryjek–Vera 2 (PRSV2-M) equation of state.

**Table 3 molecules-29-04867-t003:** Performance and stability comparison of Ni, Ni-Cu, Cu, and Ru supported on Al₂O₃ and the impact of incorporating ZnO in stabilizing the catalyst in APR.

Entry	Catalyst	Reaction Condition	H_2_ Prod. Rate (mmol/g-cat/h)	Stability
1 ^[a]^	ZnO/Ni-8Cu/Al_2_O_3_ ^[b]^ (10 wt.% Ni, 8 wt.% Cu)	Batch, 5 bar, 1 g catalyst, 100 mL of 10 wt.% **methanol**, 4 h reaction	21.8 (at 250 °C)	35% loss after 72 h
2 ^[a]^	Ni-8Cu/Al_2_O_3_ ^[c]^ (10 wt.% Ni, 8 wt.% Cu)		21.3 (at 250 °C)	75% loss after 72 h
3 ^[a]^	Ni/γ-Al_2_O_3_ (10 wt.% Ni)		10.8 (at 250 °C)	N/A
4 ^[a]^	Cu/γ-Al_2_O_3_ (10 wt.% Cu)		8.5 (at 250 °C)	N/A
5 ^[d]^	Ru/γ-Al_2_O_3_ ^[e]^ (3.7 wt.% Ru)	Fixed bed, 24 bar, 4 g catalyst, 3.6 mL/h of 10 wt.% **polyols**, WHSV = 0.3 h^−1^	Glycerol: 3.5 (at 225 °C)	18% loss after 28 h TOS
			Sorbitol: 3.3 (at 225 °C)	24% loss after 28 h TOS
			Xylitol: 2.5 (at 225 °C)	22% loss after 28 h TOS

^[a]^ Data from Liu et al. [45]. ^[b]^ Ni particle size (Transmission Electron Microscopy) = 6.1 nm. S_BET_ = 87 m^2^/g. Pore Vol. = 0.55 cm^3^/g, Pore Diameter = 27 nm. ^[c]^ S_BET_ = 114 m^2^/g. Pore Vol. = 0.78 cm^3^/g, Pore Diameter = 26 nm. ^[d]^ Data from Kalekar and Vaidya. [46]. ^[e]^ S_BET_ = 210 m^2^/g. Pore Vol. = 0.63 cm^3^/g, Pore Diameter = 13 nm.

**Table 4 molecules-29-04867-t004:** Cobalt Aluminate as a support or catalyst in the APR of methanol or glycerol and methanol.

Entry	Catalyst	Reaction Condition	H_2_ Prod. Rate (mmol/g-cat/h)	Stability
1 ^[a]^	0.625CoAl-600 ^[b]^	Fixed bed, 50 bar, 0.5 g catalyst, 10 wt.% **glycerol** in water WHSV = 24.5 h^−1^	14 (at 260 °C)	48% loss in H_2_ production rate after 30 h TOS (8 mmol-H_2_/g-cat/h)
2 ^[c]^	Pt/0.625CoAl ^[d]^ (0.3 wt.% Pt)	Fixed bed, 50 bar, 1.8 g catalyst, 10 wt.% **glycerol** in water 0.02 mL/min WHSV = 0.68 h^−1^	3.4 (at 260 °C, TOS: 10 h) 4.1 (at 260 °C, TOS: 100 h)	No decrease in glycerol conversion after 100 h TOS. H_2_ selectivity decreases from 53% to 49%.
3 ^[e]^		0.5 mL/min WHSV = 17 h^−1^	19 (at 260 °C, TOS: 3 h)	No data
4 ^[f]^	Pt/Co2Al-c700 ^[g]^ (0.98 wt.% Pt)	Batch, 20 bar, 0.1 g catalyst, 15 mL of 37 wt.% **methanol** in water, 1 h reaction	202 (at 220 °C)	9% loss in H_2_ production rate after 10 cycles of one hour each

WHSV is the weighted hour space velocity in g-feedstock/g-cat/h. ^[a]^ Data from Reynoso et al. [53]. ^[b]^ Co(FCC) particle size = 11.6 nm. S_BET_ = 101.7 m^2^/g. ^[c]^ Data from Reynoso et al. [54]. ^[d]^ Pt dispersion = 58%. S_BET_ = 131 m^2^/g. Pore Vol. = 0.52 cm^3^/g, Pore Diameter = 14.8 nm. ^[e]^ Data from Reynoso et al. [55]. ^[f]^ Data from Lv et al. [41]. ^[g]^ Pt particle size = 1.5 nm. Pt dispersion of CO chemisorption = 85%. S_BET_ = 41.5 m^2^/g. Pore Vol. = 0.19 cm^3^/g, Pore Diameter = 17.9 nm.

**Table 5 molecules-29-04867-t005:** Nickel Aluminates as Catalysts in the APR of alcohols.

Entry	Catalyst	Reaction Condition	H_2_ Prod. Rate (mmol/g-cat/h)	Stability
1 ^[a]^	NiAl_2_O_4_ ^[b]^ (33 wt.% Ni)	Fixed bed, 35 bar, 0.5 g catalyst, 10 wt.% **glycerol** in water 0.2 mL/min, WHSV = 24.5 h^−1^	26.2 (at 235 °C)	12% loss in H_2_ production rate (to 23) after 50 h TOS.
2 ^[c]^	Pt/NiAl_2_O_4_ ^[d]^ (0.97 wt.% Pt)	Fixed bed, 29 bar, 1 g catalyst, 0.05 mL/min of 10 wt.% **methanol** in water. WHSV = 2.94 h^−1^	26.4 (at 210 °C)	10% loss in conversion to gases after 600 h on stream

WHSV is the weighted hour space velocity in g-feedstock/g-cat/h. ^[a]^ Data from Morales-Marín et al. [56]. ^[b]^ Reduced at 850 °C. Ni particle size = 11.6 nm. S_BET_ = 76.6 m^2^/g. ^[c]^ Data from Li et al. [39]. ^[d]^ Pt dispersion from CO chemisorption = 80%. CO chemisorption = 40 µmol/g. H_2_ chemisorption = 233 µmol/g. S_BET_ = 147 m^2^/g.

**Table 6 molecules-29-04867-t006:** CeO_2_-supported Ni/Cu catalyst in the APR of glycerol from biodiesel byproducts.

Entry	Catalyst	Reaction Condition	H_2_ Prod. Rate (mmol/g-cat/h)	Gas Selectivity	Stability
1 ^[a]^	Ni/mp-CeO_2_ ^[b]^	Batch, 6 bar, 0.4 g catalyst 150 mL of 30 wt.% **glycerol** from biodiesel byproduct in water 2 h reaction	7.5 (at 225 °C)	75.65% H_2_ 16.87% CO_2_ 1.43% CH_4_ 6.05% CO	N/A
2 ^[a]^	1Ni-2Cu/CeO_2_ ^[c]^ (12.2 wt.%Ni, 23.3 wt.% Cu)		10 (at 225 °C)	82.72% H_2_ 14.41% CO_2_ 0.12% CH_4_ 2.74% CO	14% loss in H_2_ production rate (to 8), 10% increase in CO_2,_ and 6% in CO content after 50 cycles of 2 h each
3 ^[a]^	1Ni-2Cu/CeO_2_ + 0.2 g CaO		18 (at 225 °C)	85.08% H_2_ 14.25% CO_2_ 0.06% CH_4_ 0.61% CO	N/A

^[a]^ Data from Wu et al. [61]. ^[b]^ mp-CeO_2_: mesoporous CeO_2_. S_BET_ = 76.5 m^2^/g. Pore Vol. = 0.33 cm^3^/g, Pore Diameter = 17 nm. ^[c]^ S_BET_ = 58.8 m^2^/g. Pore Vol. = 0.22 cm^3^/g, Pore Diameter = 15 nm.

**Table 7 molecules-29-04867-t007:** CeO_2_-supported Pt catalysts prepared by photochemical reduction for the APR of methanol.

Entry	Catalyst	Reaction Condition	H_2_ Prod. Rate (mmol/g-cat/h)	Stability	Wt.% Pt in Fresh Catalyst/Spent Catalyst
1 ^[a]^	PtLa/CeO_2_ ^[b]^ (1.92 wt.% Pt, 1.29 wt.% La)	Batch, Autogenous pressure, 0.2 g catalyst, 20 mL of 10 wt.% **methanol** in water 6 h reaction	30 (at 250 °C)	17% loss in hydrogen production rate (25) after 5 cycles of six hours each at 250 °C	1.92/1.27
2 ^[a]^	Pt/CeO_2_-HT ^[c]^ (1.86 wt.% Pt)		25 (at 250 °C)	87% loss in hydrogen production rate (3.1) after 5 cycles of six hours each at 250 °C	1.86/0.39

^[a]^ Data from Lu et al. [62]. ^[b]^ H_2_ chemisorption = 20.2 µmol/g. CO_2_-TPD = 18.7 µmol/g. ^[c]^ H_2_ chemisorption = 21.6 µmol/g. CO_2_-TPD = 38.8 µmol/g.

**Table 11 molecules-29-04867-t011:** Hydrotalcite-supported catalysts in APR.

Entry	Catalyst	Reaction Condition	H_2_ Prod. Rate (mmol/g-cat/h)	Stability
1 ^[a]^	5La-NiMgAl ^[b]^ (38 wt.% Ni, 5.4 wt.% La)	Batch, Autogenous pressure, 0.2 g catalyst, 20 mL of 10 wt.% **methanol** in water, 6 h reaction	41 (at 250 °C)	20% loss in H_2_ production rate (to 33) after 5 cycles of 6 h each
2 ^[a]^	0La-NiMgAl ^[c]^ (39 wt.% Ni)		36 (at 250 °C)	72% loss in H_2_ production rate (to 10) after 5 cycles of 6 h each
3 ^[d]^	NiMg ^[e]^ (23 wt.% NiO, 51.5 wt.% MgO)	Fixed bed, 35 atm, 1.25 g catalyst., 10 vol. % **glycerol** in water, 0.102 mL/min, WHSV = 5 mL g^−1^ h^−1^	4.3 (at 250 °C)	N/A
4 ^[d]^	Ni5CuMg ^[f]^ (21.9 wt.% NiO, 5.9 wt.% CuO, 47.3 wt.% MgO)		10 (at 250 °C)	No deactivation was observed after 6 h TOS

^[a]^ Data is extracted from Huang et al. [88]. ^[b]^ Ni particle size = 3.5 nm, medium basic sites = 151 µmol-CO_2_/g. ^[c]^ Ni particle size = 4.1 nm, medium basic sites = 29.8 µmol-CO_2_/g. ^[d]^ Data is extracted from Manfro et al. [89]. ^[e]^ Ni particle size = 12 nm, S_BET_ = 173 m^2^/g, V_pore_ = 0.43 cm^3^/g. ^[f]^ Ni particle size = 6.5 nm, S_BET_ =156 m^2^/g, V_pore_ = 0.54 cm^3^/g.

**Table 12 molecules-29-04867-t012:** Selected Pt/AC catalysts.

Entry	Catalyst	Reaction Condition	H_2_ Prod. Rate (mmol/g-cat/h)	Stability
1 ^[a]^	Pt/AC ^[b]^ (4.23 wt.% Pt)	Fixed bed, 32 bar, 5 mL catalyst, 10 wt.% **glycerol** in water, LHSV = 8.4 h^−1^	18 ± 1 (at 230 °C)	No significant deactivation over about 4 h TOS
2 ^[c]^	Pt/AC (7 wt.% Pt)	Fixed bed, 45 atm, 0.3 g cat., 10 wt.% **ethylene glycol** in water, 0.1 mL/min, WHSV = 2 h^−1^	38 ± 2 (at 250 °C)	No significant deactivation over about 25 h TOS
3 ^[d]^	Pt-Fe/AC ^[e]^ (3.11 wt.% Pt, 3.11 wt.% Fe)		97 ± 3 (at 250 °C)	No significant deactivation over about 90 h TOS

WHSV is the weighted hour space velocity in g-feedstock/g-cat/h. ^[a]^ Data from Wen et al. [42]. ^[b]^ Pt dispersion = 36.4%. ^[c]^ Data from Kim et al. [95]. ^[d]^ Data from Kim et al. [96]. ^[e]^ Metal dispersion = 41.3%, metal particle size = 3.2 nm, S_BET_ = 1233 m^2^/g, Pore Vol. = 0.89 cm^3^/g, Pore Diameter = 0.91 nm.

**Table 15 molecules-29-04867-t015:** Carbon-encapsulated metal oxides supported copper catalyst in methanol APR.

Entry	Catalyst	Reaction Condition	H_2_ Prod. Rate (mmol/g-cat/h)	Stability
1 ^[a]^	29 wt.% Cu/ZnO	Batch, 20 bar, 0.05 g catalyst, 23 mL of 37 wt.% **methanol** in 0.05M KOH (aq), 1.25 h	94 (at 210 °C)	81% loss (to 18) after four cycles of 1.25 h each.
2 ^[a]^	27 wt.% Cu/ZnO@NC ^[b]^		320 (at 210 °C)	17% loss (to 266) after four cycles of 1.25 h each.
3 ^[c]^	Cu/ZnO/Al_2_O_3_ (53.1 wt.% Cu)	Batch, 20 bar, 0.05 g catalyst, 20 mL of 37 wt.% **methanol**, 1 h	87 (at 210 °C)	60% loss (to 35) after five cycles of 1.25 h each
4 ^[c]^	Cu-SP/Al_2_O_3_-ZnO ^[d]^ (52.8 wt.% Cu)		221 (at 210 °C)	44% loss (to 122) after five cycles of 1.25 h each

^[a]^ Data from Zheng et al. [107]. ^[b]^ S_BET_ = 27 m^2^/g, V_pore_ = 0.04 cm^3^/g, Average pore diameter = 20.8 nm. ^[c]^ Data from Lu et al. [108]. ^[d]^ S_BET_ = 104 m^2^/g, V_pore_ = 0.27 cm^3^/g, Average pore diameter = 10.4 nm.

**Table 17 molecules-29-04867-t017:** Molybdenum-carbide or sulfide-supported catalysts.

Entry	Catalyst	Reaction Condition	H_2_ Prod. Rate (mmol/g-cat/h)	Stability
1 ^[a]^	2%Pt/α-Mo_2_C	Batch, 20 bar, 0.1 g catalyst, 50 mL of 37 wt.% **methanol** in water. 1.25 h reaction	467 (at 190 °C)	33% loss (to 313) after 11 cycles of 1.25 h. ^[b]^
2 ^[c]^	Ni/α-Mo_2_C ^[d]^ (2.2 wt.% Ni)	Batch, 20 bar, 0.1 g catalyst, 50 mL of 64 wt.% **methanol** in water.	626 (at 240 °C)	30% loss (to 438) after 10 cycles. ^[e]^
3 ^[f]^	Pt/MoS_2_ ^[g]^ (0.2 wt.% Pt)	Batch, 20 bar. 0.2 g catalyst, 15 g of 37 wt.% **methanol** in water and 0.3 g NaOH, 1 h reaction	11.5 (@ 220 °C)	24% loss after 4 cycles of 1 h.

^[a]^ Data from Lin et al. [80]. ^[b]^ Calculated based on the total turnover number of the 1st cycle and 11th cycle. ^[c]^ Data from Lin et al. [40]. ^[d]^ S_BET_ = 80 m^2^/g. ^[e]^ Calculated based on the total turnover number of the 1st cycle and 10th cycle. Reaction time cannot be found. ^[f]^ Data from Liu et al. [37]. ^[g]^ S_BET_ = 37 m^2^/g.

## Data Availability

On request, calculations to obtain specific hydrogen production rates from all cited studies are available from C.W. Kee.

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
