# Peer review of "Thermal and Sono—Aqueous Reforming of Alcohols for Sustainable Hydrogen Production"

_molecules, 2024, doi:10.3390/molecules29204867_

Round 1
Reviewer 1 Report
Comments and Suggestions for Authors
This work reviews the thermal and sono-aqueous reforming of alcohols for sustainable hydrogen production, in which various thermal reforming catalysts and ultrasonic assisted processes are thoroughly surveyed for liquid phase reforming of alcohols. Overall, this review can provide some useful references for colleagues in this field. However, this work mainly recapitulates the research findings reported in the literatures and the own insights and generalized outlook of the authors are relatively weak. So, I recommend the authors further improve the manuscripts in order to meet the publication standards. Special comments are listed as follows.
1: The framework of the discussion section is not well structured and it is suggested that the thermo-catalytic reforming catalysts and the ultrasound-assisted process should be written in two parts.
2. The authors review the performance of various types of different carrier-loaded catalysts, but the previous work of how to improve the catalyst stability for matching the review topic of the stability of the catalysts.
3. In conclusion section, it is suggested that the authors give specific measures and technological perspectives for alcohol reforming to hydrogen technology that need to be improved or optimized in terms of catalysts and processes.
Comments on the Quality of English LanguageNo specific recommendations
Reviewer 2 Report
Comments and Suggestions for Authors
In this study, the authors present a review on the catalyst of APR hydrogen production. Generally, the content of this review is rich. However, considering the existing problems in the organization and interpretation, this manuscript may be considered for publication after the following major revisions are completed. The main comments are as follows:
-
The structure and language of the first part needs to be refined. It is recommended to use the form of a total score and provide a summary of the whole article at the end of the introduction.
-
The beginning of the discussion section on page 5 should briefly introduce the following section, while the existing part is illogical. it is suggested to further refine this paragraph of language
-
In the discussion section, it is recommended to add a summary sentence before each statement to summarize the content of each paragraph.
-
Some researches on modified alumina have been mentioned in the first and second paragraph of page 8. Is it more appropriate to place them in section 2.1.2? Besides, Fig. 5 needs to be reproduced.
-
In page 13 part 2.2.2, the introduction of PEM fuel cell is too much, which needs to be reduced. Besides, in Table 8, No. 2, 3, 4 reactions are not supported by Pt-CeO2 and should not be placed in the table.
-
In page 15 part 2.3.1, it is also recommended to add a summary paragraph to describe the characteristics of the ZrO2 catalysts.
-
Is it more appropriate to change the title of 2.4 to “other metal supported catalysts”? Besides, Fig. 6 also needs to be reproduced.
-
In page 20 part 2.5, it is suggested that the results of Wang et al. 's work should be placed in 2.5.1. In addition, several reports on hydrophobicity of carbon materials can be appropriately quoted in part 2.5 to support the conclusion.
-
2.5.4 and 2.5.5 should highlight why carbon skeleton and carbon nanotube catalysts differ from activated carbon catalysts. It should focus on its specific surface area, sintering conditions, etc. If necessary, the SEM characterization of others research should be referenced for explanation.
-
In page 27 part 2.7, it is suggested that the difference between mainstream metal APR catalyst and ultrasonic hydrogen production can be compared and draw the corresponding charts.
-
Fig. 8 needs to be refined. Besides, in page 31 part 2.7.3, it is recommended to quote the mechanism diagram to vividly show the mechanism of sonolysis.
The language of this manuscript is fluent and readable, but need to be corrected for small error such as the use of "a/an or the" and so on
Reviewer 3 Report
Comments and Suggestions for Authors
Dear Authors,
After thoroughly reading your submitting manuscript, I would to suggest to you some recommendations about enhancing the quality of your review.
1st Recommendation:
The main objective of this study was to give an overview about the status of various catalytic systems (in terms of activity and stability) in the thermal-assisted aqueous phase reforming (APR) of various alcohols. Also, the status of ultrasound-assisted APR was given as an alternative way to produce hydrogen from alcohols.
Concerning the 1st part, it was difficult for me to distinguish when the authors referred to the APR of methanol over Al2O3 - based systems and when to the APR of glycerol for the same systems. I suggest to you to make it clearer for all the mentioned catalytic systems, creating additional subsections.
2nd Recommendation:
I propose to change or modify the title of subsection 2.1.2 to Cobalt aluminate catalysts.
Also, please find a better title for the subsections 2.2.2 as you are still refer to CeO2-based systems.
3rd Recommendation:
Please change to word hypotalcite with hydrotalcite in the subsection 2.4.2.
Comments on the Quality of English Language
The level of English was quite high and I suggest minor editing.
Round 2
Reviewer 2 Report
Comments and Suggestions for Authors
What I mean previously " Fig. 5 and Fig.6 need to be reproduced" is that these two figures need to be reproduced with better quality, format, colors, and so on. The author can check figures in other manuscirpt.
For other, I have no further comments, Just published after the minor comment.
Author Response
Comments 1: What I mean previously " Fig. 5 and Fig.6 need to be reproduced" is that these two figures need to be reproduced with better quality, format, colors, and so on.
Response 1: Figure 3 has been improved by using the image obtained directly from the article's website. Minor gridline has been added to Figure 5; Subscript for TiO2 in Figure 6 has been fixed; Subscript for H2 in Figure 7 has been fixed; Figure 8 horizontal axis crossed has been shifted to -30; Subscripts in Figure 9 have been fixed.